# Geometry-Aware Equivariant Attention for Scalable Nanoelectronic Property Prediction

## Abstract

All advanced nanoelectronic devices, including transistors, image sensors, and LEDs, rely on materials and interfaces scaled down to a few nanometers. At these dimensions, material properties change in nontrivial ways due to quantum confinement and atomic-level variability, creating a multi-scale modeling challenge that requires atomistic simulations for accurate prediction. However, such simulations are often prohibitively slow or intractable, making highly expensive iterative rounds of experimentation the default option. In this work we introduce EBFormer, a geometry-aware equivariant neural network that predicts electronic properties of nanostructures by jointly capturing atomistic interactions and global geometric effects, achieving orders of magnitude speed-up over state-of-the-art physical simulators while preserving high accuracy. This is accomplished through the introduction of a novel boundary cross-attention mechanism, a scalable approach to augment local graph convolution with information of the nanostructure geometry. We validate EBFormer for nanowire and nanosheet transistors - representative of the most advanced nanoelectronic architectures currently in use - and show improved in-distribution and out-of-distribution performance on both material properties and downstream device characteristics. Combined with superior asymptotic scalability and data- and parameter-efficiency, our work paves a pathway to atomistic, automated, high-throughput and predictive nanoscale design that is otherwise not available today.

## 1 Introduction

Moore's Law and the constant demand for greater performance within fixed energy budgets and space have pushed material design into the nanoscale. Key technologies such as LEDs, image sensors, phase-change memories, and transistors now rely on materials with critical dimensions of only a few nanometers. At these dimensions the wave-nature of electrons interacts with material boundaries, leading to discrete resonant effects similar to the harmonics of sound waves on a guitar string or surface of a drum Paras et al. (2023); Ekimov & Onushchenko (2023). This quantum confinement of the electronic wavefunction leads to distinct electronic properties of nanostructures compared to bulk counterparts, and has been leveraged to engineer solar cell bandgaps Nozik (2002), boost transistor currents by up to 20% Liao et al. (2022); Park et al. (2025), and dictate LED brightness and color Xia et al. (2018). Confinement effects are also spatially heterogeneous: much like the variation of nodes and antinodes across a vibrating string, electronic properties vary across nanostructures, impacting electron distribution and other material properties Jiang et al. (2008).

In addition to geometry effects, atomic-scale variation in nanostructures also leads to outsized effects in material properties. The current-conducting channel of a modern transistor may only be tens to hundreds of atoms thick Park et al. (2025). Effects such as interfacial strain or surface defects can lead to significant variation in optical Mártil et al. (1997), electrostatic Reddy et al. (1999), and transport properties Tsutsui et al. (2019), among others.

The combination of structural and atomic-level effects leads to a unique mesoscale problem in which a large number of atoms have to be simulated to jointly capture atomic structure and the variation of material properties across the nanostructure geometry. Atomistic quantum methods such as tight-

Figure 1: **Local and global structure effects in nanoscale device properties**: An example pipeline depicting inference for an SOI transistor. Transistor properties depend on the channel electronic behavior, a nanosheet whose properties in turn depend on atomic variation and nanostructure geometry.

binding or density functional theory remain state-of-the-art for high accuracy atomistic simulation of nanostructures Jancu et al. (1998); Stradi et al. (2019); Lee et al. (2021). However, the poor scaling characteristics of these methods render them impractically slow for realistic nanoelectronic devices, making highly expensive rounds of trial-and-error experimentation the default option Yuan (2022). Machine learning has recently shown significant promise in accelerating high-accuracy but computationally-expensive atomistic quantum simulation for small molecules and crystalline systems Unke et al. (2021); Batzner et al. (2022); Musaelian et al. (2023); Batatia et al. (2024); Xie & Grossman (2018); Schütt et al. (2017); Unke et al. (2021); Ward et al. (2016). Most architectures are based primarily on local graph convolution, neglecting global structural effects. Recent architectures such as DOSTransformer Lee et al. (2023), M3GNet Chen & Ong (2022), and Neural P$^3$M Wang et al. (2024b) capture system-level features of bulk unit cells, but either do not directly encode global geometry or have poor asymptotic scaling.

In this work, we present EBFormer, a neural network designed to address the mesoscale simulation challenges of nanostructures by encoding the influence of global geometry on local material properties while preserving linear scaling. EBFormer uses equivariant attention between atoms and material interfaces Fuchs et al. (2020), a physically and empirically-motivated sparse approach to propagate global geometry information through the material structure, augmenting local chemical information captured via conventional equivariant graph convolution Tan et al. (2025). We validate our approach by predicting material properties of one- and two-dimensionally confined nanostructures representative of channels in modern advanced transistors (SOI and GAAFETs Liao et al. (2022); Huang et al. (2017)), and demonstrate superior in- and out-of-distribution performance in inference of downstream device characteristics.

## 2 BACKGROUND

### 2.1 MACHINE LEARNING FOR MATERIAL SIMULATION

Machine learning in materials science has predominantly focused on learning structure–property relationships from atomistic simulations. Among various approaches, graph neural networks (GNNs) have proven effective, as they naturally represent atomic systems. Examples of such architectures include CGNN Xie & Grossman (2018), SchNET Schütt et al. (2017), ALIGNN Choudhary & DeCost (2021) and M3GNET Chen & Ong (2022), which have demonstrated strong performance on scalar property prediction of molecules and bulk crystal unit cells including total energy, atomic forces, formation energy, and elastic moduli.

Building on this success, recent models have replaced local convolution with transformer-based attention mechanisms Vaswani et al. (2017) over local neighborhoods, leading to architectures like Equiformer Liao & Smidt (2023), COMFormer Yan et al. (2024), and Matformer Yan et al. (2022) that further improve accuracy. Moreover, both GNN-based models (e.g., ALIGNN Choudhary & DeCost (2021)) and transformer-based architectures (e.g., DOSTransformer Lee et al. (2023)) have demonstrated good performance on spectral properties such as the electronic DOS and phonon DOS. However, most of the above models primarily focus on learning local chemical environments through either graph convolution or attention, and largely overlook the role of global structural

features (such as confinement, cleaving direction, and surface termination) on macroscopic properties. Architectures including M3GNet Chen & Ong (2022) incorporate system-level features such as pressure and temperature, and DOSTransformer Lee et al. (2023) employs global cross-attention between energy and atomic embeddings, but neither explicitly model global geometry effects. Approaches such as Ewald Summations or Neural P$^3$M incorporate long-range information through reciprocal space, but scale superlinearly and require tuning multiple additional hyperparameters Kosmala et al. (2023); Wang et al. (2024b). In contrast, our proposed architecture explicitly propagates geometry-induced global effects across the system using equivariant attention while preserving linear scaling and introducing only the added hyperparameter of boundary embedding size.

Finally, previous datasets have predominantly focused on molecular and bulk-crystal systems. In such systems, local message-passing approaches are ubiquitous due to the low receptive field required to collate geometry information over the entire atomic graph. JARVIS, MC2D, C2DB, and others Choudhary et al. (2023); Haastrup et al. (2018); Liu et al. (2025) include heterostructures, but primarily focus on 2D material mono- or bilayers, which have no long-range geometric structure. In this work, we introduce a dataset of geometrically-confined materials, and explicitly demonstrate the effect of confinement on spatial variation of electronic properties (LDOS and LcDOS). To the best of our knowledge, this is the first such dataset. Further visualizations of nanosheets, nanowires, and the difference between molecular, bulk crystal, and mesoscale nanostructures are included in Appendix D.1 and Figure 6. The dataset is further detailed in Section 3.2 and Appendix C.

## 2.2 Predicted Material Properties

We focus on predicting the electronic properties of density of states (DOS) and current density of states (cDOS), used in nanoelectronic characterization to capture various downstream properties ranging from optical response to conductivity Choudhary & DeCost (2021); Lee et al. (2023). In this work, we leverage these two quantities to model the ballistic current through nanosheets and nanowires, one and two-dimensionally confined nanostructures which comprise the channels of modern transistor architectures Liao & Smidt (2023); Huang et al. (2017). Ballistic current represents the current through a transistor in the absence of electron scattering, and is a complex nonequilibrium quantity that depends on both DOS and cDOS as follows Rahman et al. (2003):

$$N_{inv} = \int dE D(E) f(E - E_f) \quad v_{inj} = \int dE \frac{J_x(E)}{N_{inv}} f(E - E_f) \quad (1)$$

$$I_{inv} = q N_{inv} v_{inj} \quad (2)$$

Here, $D(E)$ denotes the DOS , $J_x(E)$ is the cDOS along the transport direction, $f$ is the Fermi-Dirac distribution, and $N_{inv}$ represents the inversion charge density inside the nanostructure. The injection velocity $v_{inj}$ is a key figure of merit for transistors representing the average velocity of electrons entering the channel. $I_{inv}$ is the final ballistic current flowing through the transistor.

To resolve the spatial distribution of electronic states within the device, we compute atomic projections of the DOS (LDOS) and cDOS (LcDOS) using a projection operator Soriano & Palacios (2014), detailed further in Appendix B. These atom-resolved projections are crucial for capturing the microscopic distribution of charge and current within the semiconductor channel, which in turn influence the electrostatics and device characteristics.

## 3 Methods

### 3.1 Model Architecture

EBFormer is designed to capture the effects of both local atomic variation and quantum confinement on local material properties. Representing local atomic variation with graph convolution has been empirically well-motivated in literature Schütt et al. (2017); Batzner et al. (2022). Numerical simulation has supported the theory of nearsightedness of electronic materials Prodan & Kohn (2005) by demonstrating exponential decay of electronic interaction terms with distance Xia et al. (2025), establishing that local convolution with a well-tuned cutoff is sufficient to capture variation in neutral materials due to atomic defects.

Replicating long-range quantum confinement effects with neural networks, however, has limited precedence for atomic systems. Physical theory establishes that resolving the distance and orienta-

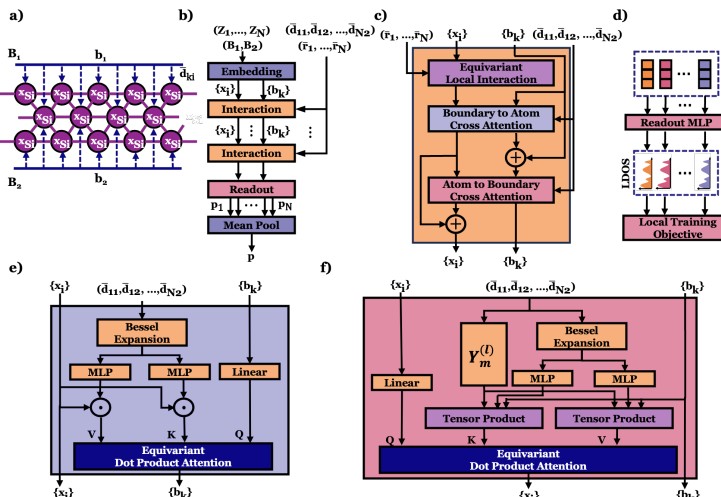

Figure 2: **EBFormer network architecture**: a) Atomistic structure with material interfaces $B_1$ and $B_2$. b) Overall architecture. c) Interaction Block containing local convolution and global attention. d) Readout neural network predicting the atomistic projection of material property. e) Equivariant attention block to update boundary nodes. f) Equivariant attention block to update atomic nodes.

tion of the boundaries relative to the crystal structure are necessary to capture their impact on local material properties Park et al. (2025); Yang et al. (2010). To accomplish this, EBFormer introduces a distance-dependent SE(3) equivariant atom-to-boundary cross-attention mechanism, enabling each atom to capture its distance to every boundary while resolving orientation through higher-order equivariant features. Our approach presents a linearly-scaling method for each atom to be aware of its local environment and its location within the nanostructure, permitting accurate prediction of local material properties. Further discussion and motivation that interaction with only boundaries is sufficient is included in Appendix Section A, where we present a comparison with an O(N²) distance-dependent equivariant self-attention mechanism, an all-to-all connected network representing a relaxation of EBFormer's boundary-only attention. We find that both architectures achieve similar errors, and that the learned self-attention matrices are sparse, focusing on local environments and atoms at the boundaries.

**Overview**   The pipeline from atomic structure to material predictions proceeds as follows. Input nanostructures are first mapped to a graph, with nodes placed at atomic positions ($\vec{r}_i$) and edges drawn between atoms within a radial cutoff. Boundary embeddings, representing interfacial planes between materials $\{\mathbf{B}_k\}$, are instantiated and initialized with a learnable vector $\mathbf{b}_k^{(0)}$, while each atomic node is initialized with an embedding $\mathbf{x}_i^{(0)}$ derived from its atomic number $Z_i$. These embeddings are iteratively refined through local graph convolution and global boundary attention layers as described below, with a final readout layer used to predict local atomic and system-level properties. Our implementation uses the E3NN library and NequIP for local convolution Geiger et al. (2022); Weiler et al. (2018); Tan et al. (2025).

**Equivariant Local Graph Convolution**   EBFormer updates atomic embeddings with local chemical information using SE(3) equivariant message-passing graph convolution introduced in NequIP Tan et al. (2025). Using an equivariant mechanism ensures preservation of the spatial symmetries of the nanostructure, which are important to ensure physical consistency of local material property predictions.

**Equivariant Global Boundary Attention**   To encode global system structure and make each atom aware of its position within the global structure, EBFormer updates local atomic embeddings using cross-attention with boundary embeddings. Boundary embeddings are also refined through cross-attention with atomic nodes, allowing them to collect global geometric information and increase

model expressivity. Both updates employ distance-aware equivariant attention mechanisms implemented with E3NN Geiger et al. (2022), as described below.

$$\mathbf{x}_i^{(t+1)} = \mathbf{x}_i^{(t)} + \sum_k \alpha_{ik} \left( \mathbf{b}_k^{(t)} \overset{\mathbf{W}_V^A}{\otimes} Y_J^{(l)}(\vec{\mathbf{d}}_{ik}) \right) \tag{3}$$

$$\mathbf{b}_k^{(t+1)} = \mathbf{b}_k^{(t)} + \sum_i \alpha_{ki} \left( \text{EquivLinear}_V^B \left( \mathbf{x}_i^{(t)} \middle| \text{MLP}_V^B (\mathbf{f}_{ik}) \right) \right) \tag{4}$$

EquivLinear $(\cdot|\mathbf{W})$ denotes an E3NN equivariant linear layer parameterized by $\mathbf{W}$, used here to incorporate the distance of atom $i$ to plane $\mathbf{B}_k$, represented as $\vec{\mathbf{d}}_{ik}$. The scalar distance is expanded over a trainable Bessel function $\mathbf{f}_{ik} = \mathbf{f}(|\vec{\mathbf{d}}_{ik}|)$, ensuring that boundary-to-atom features decay smoothly with distance so that atoms far from a boundary resemble bulk behavior. Equivariant features are combined with spherical harmonic expansions $Y_J^{(l)}$ of interatomic distance vectors using an E3NN fully-connected tensor product $\overset{\mathbf{W}}{\otimes}$ parameterized by $\mathbf{W}$. Attention scores $\alpha_{ki}$ are calculated using a softmax as follows:

$$\alpha_{ki} = \frac{\exp(\mathbf{q}_k \overset{\text{dot}}{\otimes} \mathbf{k}_i)}{\sum_j \exp \left( \mathbf{q}_k \overset{\text{dot}}{\otimes} \mathbf{k}_j \right)} \tag{5}$$

where $\overset{\text{dot}}{\otimes}$ is a parameterized fully-connected tensor product with scalar output Thomas et al. (2018). Query and keys are generated for boundary-to-atom and atom-to-boundary cross attention as:

$$\mathbf{q}_k = \text{EquivLinear}_Q^B \left( \mathbf{b}_k^{(t)} \right) \quad \mathbf{k}_i = \text{EquivLinear}_K^B \left( \mathbf{x}_i^{(t)} \middle| \text{MLP}_K^B (\mathbf{f}_{ik}) \right) \tag{6}$$

$$\mathbf{q}_i = \text{EquivLinear}_Q^A \left( \mathbf{x}_i^{(t)} \right) \quad \mathbf{k}_i = \mathbf{x}_i^{(t)} \overset{\mathbf{W}_K^A}{\otimes} Y_J^{(l)}(\mathbf{d}_{ik}) \tag{7}$$

where $\mathbf{W}_K^A = \text{MLP}_K (\mathbf{f}_{ik})$ and $\mathbf{W}_V^A$, defined similarly, are the distance-dependent weights parameterizing the fully-connected tensor product.

Note that the orientation of $\vec{\mathbf{d}}_{ik}$ and thus the $l > 0$ spherical harmonic expansions are constant for each boundary plane. This allows us to avoid the cost of a fully connected tensor product in the boundary-to-atom cross-attention (Equation 6). In contrast, atoms may interact with multiple boundary planes in different directions. Angular resolution of boundary planes is important to derive information such as cleaving planes, which has important impact on material properties Yang et al. (2010). To enable this resolution, $l > 0$ equivariance is included in the atom-to-boundary attention mechanism. The boundary mechanism is visualized for the nanosheet and nanowire cases in Appendix D.1.

**Readout** After generating the final atomic embeddings, material properties are generated per atom using a readout equivariant linear layer. For our application, we use two distinct readout heads to generate the atomic LDOS and LcDOS from the invariant features of the atomic embeddings. The system average DOS and cDOS are generated using mean-pooling on the predicted local quantities.

**Model Training** We define two loss functions $\mathcal{L}_G$ and $\mathcal{L}_L$:

$$\mathcal{L}_G = \frac{1}{N} \sum_{j,E} \mathbf{w}_{DOS} \left\| \hat{D}^j(E) - D^j(E) \right\| + \mathbf{w}_{cDOS} \left\| \hat{J}_x^j(E) - J_x^j(E) \right\| \tag{8}$$

$$\mathcal{L}_L = \frac{1}{N} \sum_{j,a,E} \mathbf{w}_{DOS} \left\| \hat{D}^j(E,a) - D^j(E,a) \right\| + \mathbf{w}_{cDOS} \left\| \hat{J}_x^j(E,a) - J_x^j(E,a) \right\| \tag{9}$$

Here, $\mathcal{L}_G$ is the mean-absolute error over total DOS and cDOS, while $\mathcal{L}_L$ is the MAE over their corresponding atomistic projections (LDOS and LcDOS). The weights $\mathbf{w}_{DOS}$ and $\mathbf{w}_{cDOS}$ balance the relative contribution of DOS and cDOS losses. EBFormer is trained on either $\mathcal{L}_G$ or $\mathcal{L}_L$, with the latter explicitly encouraging the model to capture spatial variation of material properties over the nanostructures. Further discussion and a sensitivity analysis of the relative weights of the loss terms is included in Appendix Section D. Finally, a proof of equivariance of our architecture is provided in Appendix Section J.

## 3.2 DATASET

Our dataset includes a variety of nanosheet and nanowire geometries, central building blocks of a variety of modern nanoelectronic devices Park et al. (2025); Liao & Smidt (2023); Huang et al. (2017). The nanosheet portion comprises 12,485 silicon and germanium structures, with cleaving planes varied over four orientations ($\langle 100 \rangle$, $\langle 110 \rangle$, $\langle 111 \rangle$, $\langle 211 \rangle$), nanosheet thickness scaled from 15 to 100 atomic layers, and various isotropic strains and point-defects introduced to the atomic structure. The nanowire dataset includes 380 silicon structures with each lateral dimension ranging from 10-30 atomic layers and similarly varied isotropic strains. Ground-truth labels were generated with semiempirical tight-binding using the QuantumATK simulation platform Smidstrup et al. (2019); Jancu et al. (1998). Further details regarding dataset generation, statistics, hardware used, and visualization of example structures are included in the Appendix Sections B and C. To the best of our knowledge, ours is the first dataset to capture the impact of quantum confinement on electronic properties and their spatial variation in mesoscale material systems.

## 4 EXPERIMENTS

We compare three different configurations of EBFormer on our datasets; an SE(3) invariant model trained to minimize $\mathcal{L}_G$ (EBFormer-l0), an $l = 1$ equivariant model trained to minimize $\mathcal{L}_G$ (EBFormer-l1), and an $l = 1$ equivariant model trained to minimize local loss $\mathcal{L}_L$ (L-EBFormer-l1). As an ablation of the global attention mechanism, we benchmark against NequIP-l0 and l1, using identical configuration to the local convolution mechanism of EBFormer-l0/l1. We also benchmark against DOSTransformer, a state-of-the-art model for spectral quantity prediction that incorporates global information using a standard invariant, geometry-agnostic attention mechanism. Finally, we include an MLP as a baseline. We maintain parity in the number of trainable parameters in all the models (except the ablation networks) by modifying the hidden dimension to reduce confounding effects from scaling Qu & Krishnapriyan (2024). All models are coded in PyTorch-CUDA and trained on an NVIDIA A100 GPU. Further details regarding model hyperparameters are included in Appendix Section D. Qualitative visualizations of experimental results are included in Appendix Section E.

### 4.1 IN-DISTRIBUTION NANOSHEET INFERENCE

**Nanosheet Interpolation Task** We begin by evaluating the model in the in-distribution regime. We use an 80/10/10 train/validation/test split on the entirety of the dataset, with all models trained on $\mathcal{L}_G$ with a 3:1 ratio of $\mathbf{w}_{DOS} : \mathbf{w}_{cDOS}$, except L-EBFormer-l1 that was trained on $\mathcal{L}_L$ with the same ratio. The experiment results are summarized in Table 1.

**Results** We make the following observations: **1)** Purely local models, such as the MLP and NequIP baseline approaches, show significantly worse performance in most downstream tasks compared to those that incorporate global information (DOSTransfomer and EBFormer). However, NequIP shows that incorporation of local geometric information provides a significant improvement to a geometry-agnostic MLP approach. **2)** Compared to the ablation networks, the inclusion of global structural information through the boundary nodes provides a significant improvement in inference tasks. The weight-parity experiments show that simply increasing the network size in a local approach is insufficient to reach low MAEs. **3)** Comparing EBFormer-l0 and -l1, we see that the incorporation of higher-order geometric information improved performance, even when controlling for the number of parameters. We hypothesize that this is due to the ability to resolve cleaving orientation with l1-equivariance, as discussed in Section 3.1. **4)** The more complex task of predicting local DOS/cDOS *improves* downstream mean-pooled quantity errors. We hypothesize that the

| Model | Params [M] | DOS [×10⁻³] | | cDOS [×10⁻³] | | $N_{\text{inv}}$ [×10¹¹] | | $I_{\text{inv}}$ [×10⁻³] | | $v_{\text{inj}}$ [×10⁶] | |
|---|---|---|---|---|---|---|---|---|---|---|---|
| | | MAE | RMSE | MAE | RMSE | MAE | RMSE | MAE | RMSE | MAE | RMSE |
| MLP | 9.25 | 14.82 (0.02) | 23.03 (0.27) | 12.07 (0.10) | 20.22 (0.51) | 2.69 (0.01) | 6.21 (0.04) | 7.26 (0.05) | 19.99 (0.62) | 1.63 (0.05) | 2.63 (0.08) |
| NequIP-l0 (Ablation) | 1.39 | 5.59 (0.01) | 9.14 (0.08) | 2.40 (0.05) | 4.51 (0.06) | 0.45 (0.00) | 1.35 (0.03) | 1.43 (0.03) | 4.26 (0.10) | 0.87 (0.01) | 1.97 (0.19) |
| NequIP-l1 (Ablation) | 1.18 | 5.50 (0.05) | 8.92 (0.04) | 2.31 (0.06) | 4.31 (0.11) | 0.44 (0.01) | **1.30** (0.04) | 1.36 (0.04) | 4.06 (0.10) | 0.88 (0.02) | 1.83 (0.08) |
| NequIP-l0 (Parity) | 9.98 | 5.53 (0.01) | 9.17 (0.05) | 2.29 (0.00) | 4.39 (0.04) | 0.44 (0.00) | 1.36 (0.02) | 1.37 (0.01) | 4.18 (0.03) | 0.91 (0.04) | 2.24 (0.50) |
| NequIP-l1 (Parity) | 9.46 | 5.46 (0.06) | 8.91 (0.05) | 2.30 (0.08) | 4.27 (0.07) | 0.44 (0.01) | 1.31 (0.05) | 1.36 (0.06) | 4.02 (0.10) | 0.85 (0.02) | 1.70 (0.31) |
| DOSTransformer | 9.40 | **2.80** (0.15) | 6.59 (0.12) | 1.55 (0.09) | 3.80 (0.09) | 0.37 (0.02) | 1.33 (0.04) | 0.90 (0.04) | 3.51 (0.16) | 0.80 (0.16) | 2.34 (1.39) |
| EBFormer-l0 | 9.48 | 3.27 (0.10) | 6.89 (0.07) | 1.32 (0.05) | 3.93 (0.04) | 0.33 (0.01) | 1.38 (0.01) | 0.78 (0.03) | 3.75 (0.05) | 0.61 (0.02) | 1.25 (0.14) |
| EBFormer-l1 | 9.35 | 3.17 (0.06) | 6.74 (0.14) | **1.26** (0.03) | 3.76 (0.16) | 0.32 (0.01) | 1.38 (0.05) | 0.75 (0.02) | 3.61 (0.11) | 0.60 (0.01) | 1.35 (0.28) |
| L-EBFormer-l1 | 9.35 | 2.85 (0.06) | **6.21** (0.10) | **1.26** (0.04) | **3.37** (0.11) | **0.31** (0.00) | **1.30** (0.03) | **0.67** (0.02) | **3.24** (0.13) | **0.49** (0.01) | **0.90** (0.07) |

Table 1: **Interpolation Errors** DOS (eV⁻¹), cDOS (eV⁻¹cm⁻¹s⁻¹), $N_{inv}$ (cm⁻²), $I_{inv}$ (mA/$\mu$m), and injection velocity (cm/s) errors on the interpolation task are shown below. Errors are averaged over three seeds with standard deviation in parentheses. Bold figures denote the best (lowest) mean in each sub-column.

added information of spatial variation of quantities through the structure provides useful information regarding the structural impact on LDOS/LcDOS, and therefore on the total nanostructure properties. Note that DOSTransformer is unable to predict local quantities, and therefore cannot leverage this information. **5)** EBFormer-l1 shows slightly lower performance compared to DOSTransformer in DOS prediction, but comparable or improved performance in cDOS and downstream tasks, notably in injection velocity. Injection velocity is a ratio of quantities derived from DOS and cDOS, and thus depends on not only the accuracy but the mutual consistency of the two values. High performance on this downstream tasks indicates physically consistent covariance of EBFormer's predictions, and thus a more rich learned embedding. Analysis of correlation and consistency of these quantities are included in the Supplementary Section F. **6)** L-EBFormer-l1 shows significantly improved performance on cDOS and all downstream tasks, while showing comparable performance to DOSTransformer on the DOS task. This demonstrates that incorporating geometric information in global attention improves material property prediction in structured materials.

## 4.2 OUT-OF-DISTRIBUTION NANOSHEET INFERENCE

Compared to bulk materials in which a single unit-cell fully determines system properties, nanostructures incorporate a large number of atoms with unique local environments that define the system behavior. Simulating nanostructures involves very large unit-cells, and therefore is computationally difficult. Models that are capable of capturing the behavior of much larger systems from smaller, easier to simulate structures would therefore be of great value.

| Model | Params [M] | DOS [×10⁻³] | | cDOS [×10⁻³] | | $N_{\text{inv}}$ [×10¹¹] | | $I_{\text{inv}}$ [×10⁻³] | | $v_{\text{inj}}$ [×10⁶] | |
|---|---|---|---|---|---|---|---|---|---|---|---|
| | | MAE | RMSE | MAE | RMSE | MAE | RMSE | MAE | RMSE | MAE | RMSE |
| MLP | 0.99 | 27.69 (5.88) | 41.18 (9.80) | 23.88 (1.56) | 37.09 (1.72) | 2.38 (0.28) | 5.15 (0.20) | 17.93 (0.43) | (1.72) | 7.94 (0.12) | 13.20 (1.35) |
| NequIP-l1 (Parity) | 1.04 | 8.82 (0.30) | 11.70 (0.33) | 4.39 (0.14) | 6.05 (0.30) | 1.22 (0.02) | 2.44 (0.06) | 2.68 (0.06) | 5.64 (0.34) | 2.37 (0.04) | 2.98 (0.10) |
| DOSTransformer | 0.95 | 11.88 (1.18) | 16.89 (0.79) | 7.11 (0.63) | 10.16 (0.50) | 1.42 (0.20) | 3.21 (0.20) | 4.20 (0.92) | 8.57 (1.66) | 4.99 (2.31) | 7.34 (3.61) |
| EBFormer-l0 | 0.94 | 9.12 (0.19) | 12.42 (0.17) | 4.37 (0.26) | 7.41 (0.23) | 1.35 (0.17) | 3.04 (0.17) | 2.42 (0.16) | 6.61 (0.21) | 1.00 (0.17) | 1.32 (0.14) |
| EBFormer-l1 | 0.91 | 7.97 (0.23) | 10.89 (0.25) | **3.37** (0.23) | **5.39** (0.26) | **1.00** (0.09) | **2.29** (0.10) | **1.81** (0.16) | **4.69** (0.25) | **0.89** (0.09) | **1.24** (0.05) |
| L-EBFormer-l1 | 0.91 | **7.52** (0.24) | **10.55** (0.44) | 4.58 (0.29) | 6.98 (0.62) | 1.03 (0.02) | 2.48 (0.15) | 2.76 (0.35) | 6.60 (0.87) | 0.97 (0.08) | 1.35 (0.06) |

Table 2: **Nanosheet Extrapolation Errors** DOS (eV⁻¹), cDOS (eV⁻¹cm⁻¹s⁻¹), $N_{inv}$ (cm⁻²), $I_{inv}$ (mA/$\mu$m), and injection velocity (cm/s) errors on the nanosheet extrapolation task (training on 15-45 layers, test on 46-100 layers) are shown below. Errors are averaged over three seeds with standard deviation in parentheses. Bold figures denote the best (lowest) mean in each sub-column.

**Nanosheet Extrapolation Task** To quantify extrapolative performance in the size of the system, we train the model on silicon nanosheets from 15 to 45 atomic layers thick with varied strain, cleav-

ing, and point-defects. We test and validate on nanosheets from 46-100 layers. All test errors are generated using the model with the minimum validation loss during training.

**Results** Errors on the extrapolation task are shown in Table 4.2. We notice first that the weight-parity local-convolution network is more robust to the distribution shift than DOSTransformer. This is likely because the GNN is unaware of the global structure, and therefore predicts similar local properties compared to smaller structures, which are good first-order approximations. DOSTransformer includes global information with geometry-agnostic attention, and therefore therefore cannot leverage physical information regarding variation of properties due to geometry well. EBFormer demonstrates that our mechanism of symmetry-aware inclusion of geometry in global attention is better able to capture the physics of quantum confinement for application to out-of-distribution situations. Finally, we notice that L-EBFormer generally performs more poorly than the global prediction approach. We hypothesize that this is due to the added complexity and variability of the local quantities compared to the system-level DOS and cDOS, which may lead to decreased robustness under distribution shift.

While there is no standard metric for required accuracy, architectural decisions between nodes typically yield 15-60% impact in quantities such as power, current, and electron mobility Park et al. (2025). Resolving the impact of these decisions requires less than 10% errors in predictions. DOSTransformer, the strongest model after EBFormer, demonstrated 17% nRMSE in-distribution and 52% in out-of-distribution nanosheet inference of injection velocity, while EBFormer demonstrated errors of 6% in-distribution and 9% in the OOD case (statistics from Appendix C).

## 4.3 IN-DISTRIBUTION NANOWIRE INFERENCE

Finally, we benchmark EBFormer against other architectures on the nanowire dataset. This system introduces an additional dimension of confinement, leading to distinct physical effects compared to the nanosheet case Huang et al. (2017); Jiang et al. (2008). We adopt the same 80/10/10 train/validation/test split and loss functions as in the nanosheet interpolation experiment. Errors are summarized in Table 3.

| Model | Params [M] | DOS [×10⁻³] | | cDOS [×10⁻³] | | $N_{inv}$ [×10¹¹] | | $I_{inv}$ [×10⁻³] | | $v_{inj}$ [×10⁶] | |
|---|---|---|---|---|---|---|---|---|---|---|---|
| | | MAE | RMSE | MAE | RMSE | MAE | RMSE | MAE | RMSE | MAE | RMSE |
| MLP | 0.99 | 22.87 | 35.81 | 9.66 | 12.88 | 0.23 | 0.60 | 5.45 | 11.23 | 3.27 | 4.98 |
| | | (0.19) | (0.58) | (1.42) | (1.64) | (0.02) | (0.06) | (1.25) | (2.00) | (0.77) | (1.42) |
| NequIP-l1 (Parity) | 1.04 | 18.12 | 31.68 | 2.97 | 4.39 | 0.11 | 0.24 | 0.89 | 1.97 | 1.67 | 2.12 |
| | | (0.12) | (0.19) | (0.25) | (0.28) | (0.01) | (0.01) | (0.28) | (0.59) | (0.04) | (0.06) |
| DOSTransformer | 0.95 | 22.20 | 33.57 | 5.61 | 7.53 | 0.14 | 0.30 | 2.38 | 4.15 | 2.98 | 3.58 |
| | | (0.65) | (0.90) | (0.75) | (0.64) | (0.04) | (0.07) | (1.43) | (2.14) | (1.91) | (2.17) |
| EBFormer-l0 | 0.94 | 14.87 | 27.95 | 1.20 | 2.19 | 0.08 | 0.17 | 0.29 | 0.74 | 1.55 | 1.94 |
| | | (0.54) | (0.29) | (0.12) | (0.18) | (0.00) | (0.00) | (0.05) | (0.13) | (0.02) | (0.01) |
| EBFormer-l1 | 0.91 | 13.55 | 26.72 | **0.95** | **1.76** | **0.08** | **0.17** | **0.26** | **0.67** | 1.51 | 1.90 |
| | | (0.21) | (0.24) | (0.08) | (0.28) | (0.00) | (0.00) | (0.02) | (0.15) | (0.02) | (0.04) |
| L-EBFormer-l1 | 0.91 | **12.94** | **26.45** | 1.26 | 2.17 | 0.08 | 0.17 | 0.34 | 0.77 | **1.47** | **1.84** |
| | | (0.01) | (0.07) | (0.06) | (0.10) | (0.00) | (0.00) | (0.07) | (0.14) | (0.02) | (0.01) |

Table 3: Prediction accuracy (MAE and RMSE) for the nanowire interpolation task. Errors are averaged over three seeds with standard deviation in parentheses. DOS (eV$^{-1}$), cDOS (eV$^{-1}$cm$^{-1}$s$^{-1}$), $N_{inv}$ (cm$^{-2}$), $I_{inv}$ (mA/$\mu$m), and injection velocity (cm/s) are scaled as indicated for readability. Bold figures denote the best (lowest) mean in each sub-column.

We first observe that the local convolution network outperforms DOSTransformer despite having no nonlocal structural information. We attribute this to the flexible inductive bias of DOSTransformer relative to NequIP, which in this data-sparse regime likely leads to greater model variance and overfitting. In contrast, EBFormer generalizes well to the nanowire task, achieving significantly better performance than all other architectures. We also note, however, that L-EBFormer again underperforms relative to its global variant. We hypothesize that this is due to the increased complexity of modeling the two-dimensional variation of LDOS and LcDOS across the nanowire, a harder task compared to the one-dimensional nanosheet which makes local material property inference challenging.

## 4.4 MODEL PARAMETERIZATION AND LEARNING CURVES

Figure 3 shows learning curves and parameter efficiency of EBFormer compared to NequIP and DOSTransformer on the nanosheet interpolation task. We notice that EBFormer achieves low losses

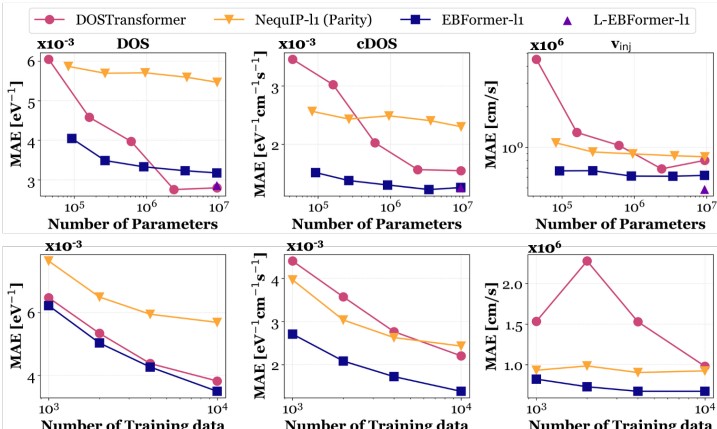

Figure 3: **Parameterization and Learning Curves**: Various model configurations are tested on the nanosheet interpolation task. Learning curves models maintain weight parity at $\sim$ 1M parameters. We observe superior performance with EBFormer in the low-data regime, and observe both that EBFormer performance continues to scale with increased parameter count while DOSTransformer plateaus, as well as superior performance in the low parameter regime, indicating the suitability of the inductive bias.

in cDOS and injection velocity with a relatively small number of parameters, implying the model inductive bias is well-suited to the inference task. We also see that training on local projections of DOS and cDOS (L-EBFormer-l1) shows marked improvement in predictions with the same number of model parameters. While DOSTransformer achieves superior DOS losses at high parameterization, we notice that EBFormer continues to improve with scaling, and achieves superior downstream injection velocity performance.

Improved injection velocity performance is also apparent in the learning curves; we see that EBFormer achieves better injection velocity losses with 1k training data than DOSTransformer with 10k. In contrast, NequIP, a purely local convolutional model, saturates with few data. By augmenting local convolution with global structural context, EBFormer demonstrates a significant shift in learning efficiency and predictive accuracy, surpassing both NequIP and DOSTransformer.

## 5 CONCLUSION

In this work, we propose EBFormer, a physically-motivated architecture for atomistic material property prediction in confined nanoscale systems. Through equivariant geometry-aware cross attention between atom nodes and boundary nodes representing interfacial planes, we demonstrate high performance in in-distribution atomistic electronic property predictions and downstream system-level tasks in one- and two-dimensionally confined nanostructures. We also show the reproduction of geometry-induced effects in electronic properties – which local atomistic approaches fail to capture – and demonstrate high-fidelity reproduction of spatial variation of properties within systems with different dimensionalities. Finally, we show superlative performance compared to state-of-the-art models in the data-limited and extrapolative regimes, demonstrating the strength of our model's

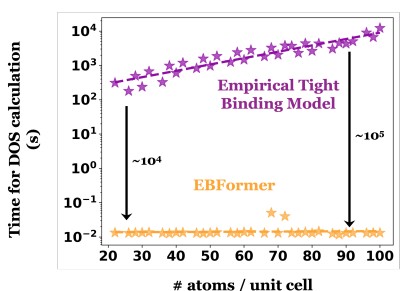

Figure 4: **Acceleration Over Physical Simulation**: Details in Appendix G

inductive bias. EBFormer preserves linear scaling in system size, and achieves 4–5 orders of magnitude acceleration over conventional atomistic simulations (Figure 4). Through a combination of scalability, predictive accuracy, and the capacity to model geometric effects at the nanoscale regime, EBFormer presents a concrete framework for the rapid design and modeling of next-generation nanoscale materials and devices.

## 6 ETHICS STATEMENT

We acknowledge that we have carefully reviewed the ICLR ethics statement, and assure that all studies in this work adhere to the outlined guidelines. EBFormer is a neural network architecture designed to capture quantum confinement effects in nanostructures. Nanostructures are prevalent across various domains, including materials science and semiconductor device engineering. By enabling accelerated and accurate modeling of these systems, EBFormer can significantly enhance the design cycle for next-generation, energy-efficient devices. This has the potential to positively impact society by promoting sustainable technologies. However, training such large neural networks, particularly those incorporating equivariant features and attention operations, can be very computationally intensive and energy demanding. As global energy consumption from machine learning continues to rise, such models risk contributing to environmental degradation if not paired with sustainable computing practices.

## 7 REPRODUCIBILITY

The code is available at https://github.com/EBFormer/EBFormer, containing the neural network configuration files, training scripts, and test scripts to reproduce our results. Appendix sections B, C, and D also contain the dataset splits and the implementation details about the neural network architecture. The datasets for both interpolation tasks and the nanosheet extrapolation tasks are available at Zenodo .

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

APPENDIX

## A ARCHITECTURAL MOTIVATION

Our work presents a generalizable framework to incorporate the effects of atomic interactions with boundary planes on local and global electronic structural variation. Specifically, we focus on structured materials with long-range order such as those dictating the electronic properties of modern nanoelectronic devices, which can be comprised of hundreds of thousands of atoms. In such contexts, favorable asymptotic scaling is essential, and warrants careful engineering of the model inductive bias. The novelty of our approach lies in explicitly introducing boundaries into atomic graph networks, and introducing a framework to communicate requisite long-range information to and from atomic nodes via equivariant attention mechanisms to capture geometric confinement. Crucially, we do this in a manner preserving O(N) asymptotic scaling, and demonstrate high accuracy and extensibility to nanosheets and nanowires, including in OOD situations. In the following paragraphs, we motivate the main architectural decisions of EBFormer as both necessary and sufficient, and emphasize the suitability of its inductive bias to our task of modeling atomic interactions with boundaries.

**Necessity of our Architectural Components** The local electronic properties of an atom in a nanostructure depend on the local atomic environment as well as the global material structure. For example, the local density of states in a crystalline nanosheet is determined by local properties such as the system lattice constant, elemental composition of the unit-cell, local atomic defects, and others. To capture such local effects, we motivated the utilization of a GNN with a local cutoff.

In addition to local effects, nanostructure critical dimensions can be significantly smaller than electronic de Broglie wavelengths and phase decay lengths, which leads to quantum confinement Abe et al. (2022). To capture the long-range information of confinement, we introduced boundary nodes, vector embeddings to collect and communicate information regarding geometric confinement. We utilized a cross-attention mechanism to exchange the heterogeneous information between the boundary embeddings and atom embedding while incorporating atom-boundary distance, a crucial factor that determines confinement. Furthermore, we utilize Bessel expansions for over the boundary-atom distance, ensuring that interactions asymptotically decay as boundaries move far from atoms, leading local material properties to approach bulk behavior. In our main text, we showed that communication with the boundary planes was necessary to achieve high accuracy with our ablations with NequIP, the underlying local GNN. Furthermore, we demonstrated the necessity of an equivariant interaction with the boundary planes through another ablation with EBFormer-l0, which enables encoding angular information to determine cleaving direction (the relative orientation of the boundary plane with the crystal structure).

**Sufficiency of our Architectural Components** Limiting attention to only boundaries, however, is a strong inductive bias. Currently, we only include local and boundary information. It is possible that this is insufficient, and information about all atoms in the structure are required to be communicated (for example with a mechanism such as in Neural $P^3M$ Wang et al. (2024b)). To answer this question, we relaxed the inductive bias of boundary-only attention to permit all atom-to-atom interactions, replacing the boundary interaction step with a fully trainable, equivariant, distance-conditioned self-attention mechanism identical to the boundary node interaction mechanism but between all atoms instead. In this way, we could visualize the attention matrices between atoms to determine if the interatomic interactions are dense or have some pattern or sparsity.

| Model | Params (M) | DOS MAE | DOS RMSE | cDOS MAE | cDOS RMSE |
|---|---|---|---|---|---|
| NequIP-l1 (Parity) | 0.27 | 5.69 | 9.19 | 2.43 | 4.48 |
| DOSTransformer | 0.27 | 3.82 | 7.58 | 2.20 | 4.63 |
| Self-Attention-l1 | 0.27 | 3.67 | 6.96 | **1.34** | **3.33** |
| EBFormer-l1 | 0.27 | **3.49** | **6.94** | 1.38 | 3.77 |

Table 4: Model size and errors for DOS and cDOS for the self-attention network compared to NequIP (which doesn't include nonlocality), DOSTransfomer (nonlocality without geometry information), and EBFormer, which only includes boundary attention

Table 4 presents the errors of the self-attention architecture on the interpolation nanosheet task compared to the purely local NequIP, the cross-attention globally-aware mechanism of DOSTransformer, and EBFormer-l1, which uses boundary attention. We note that EBFormer and the self-attention network are significantly superior to the other two methods, while both achieve similar performance to the other, providing motivation that the mechanism we use is sufficient. This is reinforced by the attention masks (visualized in Figure 5). We utilized four layers of self-attention for the nanosheet, identical to EBFormer, and present a the top 5% of attention scores sorted by score below for a 97-layer silicon nanosheet cleaved along $\langle 100 \rangle$. Atomic indices are ordered by z-axis position, from least to most, along the nanosheet. Atoms 0-1 and 98-99 are passivating hydrogens masked when predicting total system electronic properties. For the nanowire, due to the larger size of the structures, we could only fit three layers of self-attention within our GPU, visualized in panel b) for a 13x13 silicon nanowire cleaved along $\langle 100 \rangle$. The architecture is highlighted in panel c).

Visually, the attention mask for the nanosheet has the form of an "N", with a diagonal component (which is local), and two strong nonlocal "legs" which show that each atom has a high attention score with at least one of the atoms at the boundaries. Masks 2-4 primarily demonstrate local attention, with strong scores mainly along the diagonal. Similarly, the nanowire attention scores

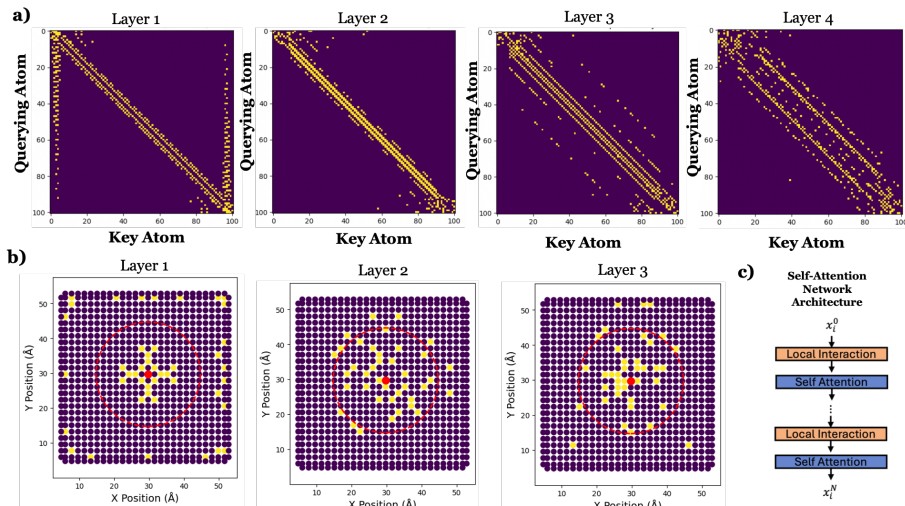

Figure 5: **Visualization of the Attention Matrices**: (a) Top 5% off attention scores (four layers of self-attention) for a silicon nanosheet with 97 layers. The attention pattern is sparse, with a strong local (on-diagonal) component, and nonlocal "legs" representing attention with the boundaries. (b) Top 5% of attention scores (three layers of self-attention) of the central atom in a 26x26 layer silicon nanowire (represented in real space as the cross-section of the nanowire, where dots are atoms). Red circle represents the convolutional layer receptive field. Attention scores appear to focus locally (within the receptive field of the convolution) or the boundaries. (c) Architecture of the self-attention network. Interperses local graph convolution with nonlocal, fully-connected self-attention using an identical attention formulation as EBFormer

(visualized over the 2D face of the nanowire for the central atom) show that top 5% scores are either on the boundaries (in layer 1), or primarily within the receptive field (layers 2-3). Leveraging the sparsity of these attention matrices, we motivate that including nonlocal interactions only with the boundaries of a structure (representing the "legs" of the "N" for the nanosheet, or the boundary scores in the nanowire) is *sufficient* for our systems of interest. Rather than connecting all atoms with all surface atoms, we use a single vector embedding to collect boundary information, which ensures preservation of O(N) scaling for arbitrary geometries. Finally, augmenting the nonlocal information with local convolution covers the diagonal of the "N" in the attention mask of the nanosheet, or the scores within the circle of the nanowire.

With the preceding discussion, we have motivated the architectural approaches taken with EBFormer as required and physically-motivated but frugal, with only the requisite interactions included to preserve favorable asymptotic scaling.

**Empirical Validation**   To conclude, we note that our architecture demonstrates superior in-distribution and OOD when compared to a global, linearly scaling approach DOSTransformer which does not carefully consider the inductive biases of our domain. Furthermore, we show the suitability of the inductive bias of our approach by demonstrating high performance with parameter and data sparsity. We also demonstrate the improvement of our predictions in the OOD thickness case with inclusion of larger structures from another elemental species in the transfer learning task in Appendix I, demonstrating the ability of our architecture to leverage structure information separately from local information. Finally, we also demonstrate our method's applicability to nanowires such as those in GAAFETs which face an additional dimension of confinement. These results serve to further empirically motivate that the EBFormer architecture is well-suited to our task of capturing boundary effects with efficient scaling and preservation of a local basis.

We note that the matrices from the self-attention experiment presented above were our initial motivation to engineer EBFormer.

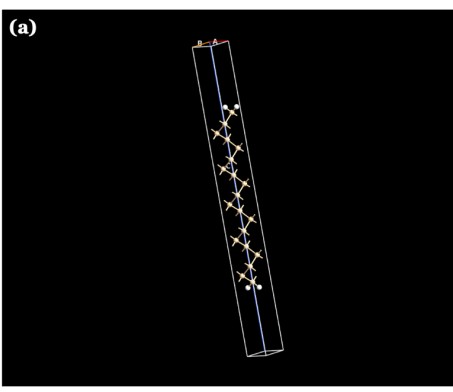 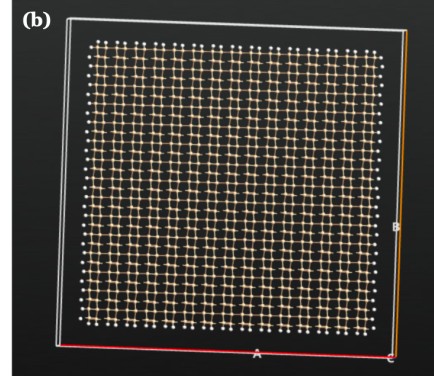

Figure 6: **Visualization of the Structures**: (a) Silicon $\langle 100 \rangle$ nanosheet visualization (b) Silicon 30x30 layer nanowire visualization. Tan atoms are silicon, and the white atoms are hydrogen.

Table 5: Counts of cleaving directions for silicon and germanium nanosheets

| Material | $\langle 1\,0\,0 \rangle$ | $\langle 1\,1\,0 \rangle$ | $\langle 1\,1\,1 \rangle$ | $\langle 2\,1\,1 \rangle$ | Total |
|---|---|---|---|---|---|
| Silicon | 3,786 | 2,353 | 1,126 | 481 | 7,746 |
| Germanium | 2,458 | 1,394 | 444 | 443 | 4,739 |
| Total | 6,244 | 3747 | 1570 | 924 | 12,485 |

# B  DATA GENERATION

Our nanosheet dataset consists of 12,485 confined, hydrogen-passivated nanosheets representing the channels of SOI transistors Park et al. (2025). Table 5 shows the distribution over cleaving orientation and elemetal species. The nanowire dataset consists of 380 confined, hydrogen-passivated nanowires representing the channles of GAAFAETs Huang et al. (2017). Density of states and current density of states Rahman et al. (2003) are derived for each structure using an empirical tight-binding simulation Jancu et al. (1998). Our structure generation and simulation are both conducted using QuantumATK Smidstrup et al. (2019), with details presented in the following sections. All simulations were conducted on Intel Xenon Gold 6330 processors parallelized across 16 to 64 processes.

## B.1  STRUCTURE GENERATION

We begin with a unit-cell of either silicon or germanium material in the face-centered diamond cubic crystal configuration. The system lattice constant is scaled by a multiplicative constant in the range of (0.9, 1.035) for silicon, and (0.9, 1.02) for germanium to simulate the effect of strain. The strained unit cell is then cleaved to form a nanosheet such that the normal direction of the sheet is oriented along one of four orientations: $\langle 100 \rangle$, $\langle 110 \rangle$, $\langle 111 \rangle$, or $\langle 211 \rangle$, or to form a nanowire, with orientation selected along $\langle 100 \rangle$. Final nanosheet atomic layer count is varied between 15 to 100 layers for all cleaving directions, or 10-30 layers in each lateral direction for the nanowire. Point-defects are introduced for a subset of silicon nanosheet structures, with displacement amplitude chosen in the range of (0.05 Å, 0.15 Å). Finally, the system is passivated with hydrogen using the QuantumATK passivation tool to reduce surface defects due to dangling bonds. The displacement amplitude and strain ranges were chosen to ensure preservation of a band gap to retain the semiconductor nature of the nanostructures.

## B.2  ELECTRONIC STRUCTURE SIMULATION

Using the generated atomic structures, electronic properties are derived using an empirical tight-binding simulation (commonly used for atomistic property simulation for semiconductors Vogl et al. (1983)) with QuantumATK. Silicon structures are simulated using the Bassani.SiH basis set, while

germanium-containing structures use Bassani.GeH Jancu et al. (1998). LDOS is generated using tetrahedral expansion over energies calculated at a 150x150 Monkhorst-Pack grid over $k_x$ and $k_y$. cDOS is calculated using Gaussian broadening of 0.01 eV over the band structure sampled over the same grid. Projection for LDOS and LcDOS calculation is conducted using a projection operator $\hat{\mathbf{P}}_a$ onto atom $a$ Soriano & Palacios (2014):

$$D(E, a) = \sum_{\vec{k}} \delta(E - E_{\vec{k}}) \langle \psi_{\vec{k}} | \hat{\mathbf{P}}_a | \psi_{\vec{k}} \rangle; \quad J_x(E, a) = J_x(E) \times \frac{D(E, a)}{D(E)} \qquad (10)$$

Both DOS/LDOS and cDOS/LcDOS are calculated over 100 energy bins from $E_c$ - 0.01 eV to $E_c$ + 0.3 eV, where $E_c$ is the conduction band minimum. Injection velocity and inversion charge and current calculations are conducted by varying the fermi level over 100 energies in the range (-0.3 eV, 0.3 eV) around $E_c$ Rahman et al. (2003).

## C    DESCRIPTION OF DATASETS

### C.1    IN-DISTRIBUTION NANOSHEET DATASET

The statistics and distributions of quantities in the dataset for the nanosheet interpolation task are presented below. The task uses a 80-10-10 randomized split over all structures in the dataset. Figure 7 depicts the distributions of DOS, cDOS, and injection velocity values across all energy bins and fermi levels in the test, training, and validation sets. Units and scales are the same as presented in all error tables.

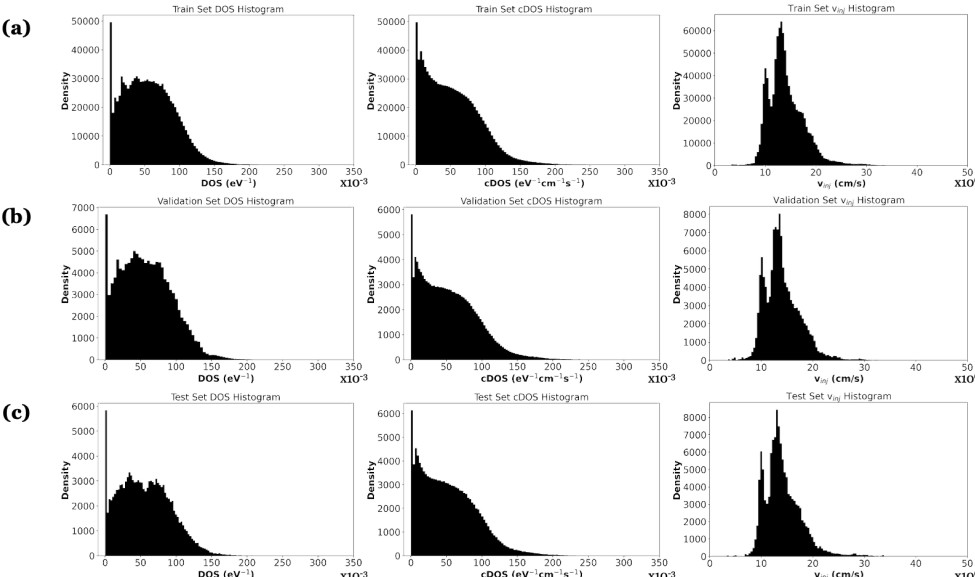

Figure 7: **Statistics of the In-Distribution Nanosheet Dataset**: (a) Training Data Histograms for DOS, cDOS, and injection velocity (b) Validation Data Histograms (c) Test Data Histograms

Table 6 includes the means and standard deviations across all energy bins and fermi levels of DOS, cDOS, and downstream derived quantities over the test, train, and validation datasets. From the histograms and tables, we can draw the conclusion that the training data is representative of the test task.

### C.2    OUT-OF-DISTRIBUTION NANOSHEET DATASET

The statistics and distributions of quantities in the dataset for the zero-shot out-of-distribution task are presented below. The extrapolation task uses nanosheets 15-45 atomic layers thick of all strains, cleaving directions, and point-defective silicon nanosheets as the training data, and validates/tests on

Table 6: **In-Distribution nanosheet statistics**: DOS ($eV^{-1}$), cDOS ($eV^{-1}cm^{-1}s^{-1}$), $N_{inv}$ ($cm^{-2}$), $I_{inv}$ (mA/$\mu$m), and injection velocity (cm/s) means over the train, validation, and test set for the in-distribution task are included below. Standard deviations are presented in parentheses below

| Set | Count | DOS $[\times 10^{-3}]$ | cDOS $[\times 10^{-3}]$ | $N_{inv}[\times 10^{11}]$ | $I_{inv}[\times 10^{-3}]$ | $v_{inj}[\times 10^{6}]$ |
|---|---|---|---|---|---|---|
| Train Stats | 9991 | 56.35 (35.75) | 52.80 (38.60) | 10.29 (15.98) | 28.67 (47.08) | 14.10 (3.60) |
| Validation Stats | 1246 | 57.52 (36.21) | 54.10 (39.55) | 10.49 (16.22) | 29.48 (48.56) | 14.06 (3.49) |
| Test Stats | 1248 | 56.75 (35.65) | 53.46 (39.04) | 10.32 (15.98) | 29.11 (48.00) | 14.12 (3.58) |

46-100 layer structures. The validation and tests datasets are a 50-50 randomized distribution of the 46-100 layer nanosheets. Figure 8 depicts the distributions of DOS, cDOS, and injection velocity values across all energy bins and fermi levels in the test, training, and validation sets. Units and scales are the same as presented in all error tables.

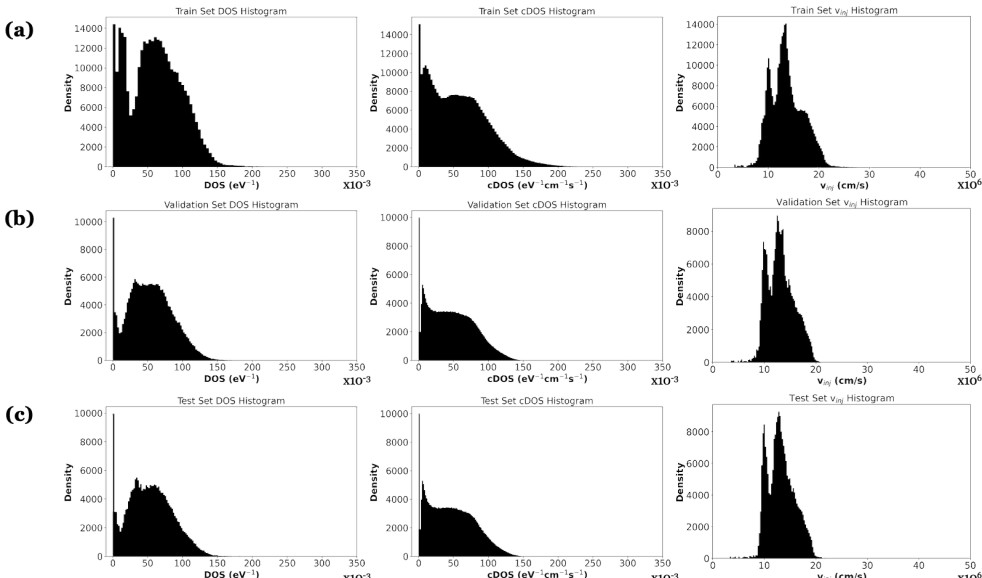

Figure 8: **Statistics of the Out-of-Distribution Nanosheet Dataset**: (a) Training Data Histograms for DOS, cDOS, and injection velocity. These are silicon nanosheets with atomic layers from 15-45 inclusive. (b) Validation Data Histograms (c) Test Data Histograms. Validation and test set are over silicon structure with 46-100 layers inclusive. A distribution shift is apparent between the training and test data

Table 7: **Out-of-Distribution nanosheet statistics**: DOS ($eV^{-1}$), cDOS ($eV^{-1}cm^{-1}s^{-1}$), $N_{inv}$ ($cm^{-2}$), $I_{inv}$ (mA/$\mu$m), and injection velocity (cm/s) means over the train, validation, and test set for the out-of-distribution task are included below. Standard deviations are presented in parentheses below

| Set | Count | DOS $[\times 10^{-3}]$ | cDOS $[\times 10^{-3}]$ | $N_{inv}[\times 10^{11}]$ | $I_{inv}[\times 10^{-3}]$ | $v_{inj}[\times 10^{6}]$ |
|---|---|---|---|---|---|---|
| Train Stats | 3291 | 60.24 (37.03) | 57.65 (41.24) | 11.42 (17.23) | 32.63 (54.60) | 13.62 (3.25) |
| Validation Stats | 2225 | 53.79 (31.41) | 47.44 (32.25) | 10.00 (15.24) | 25.45 (39.75) | 13.19 (2.64) |
| Test Stats | 2227 | 53.88 (31.49) | 47.53 (32.30) | 10.06 (15.35) | 25.49 (39.81) | 13.24 (2.61) |

Table 7 includes the means and standard deviations across all energy bins and fermi levels of DOS, cDOS, and downstream derived quantities over the test, train, and validation datasets for the extrapolation task. We qualitatively see from the histograms that the training and test distributions are dissimilar.

## C.3    In-Distribution Nanowire Experiment

The statistics and distributions of quantities in the dataset for the nanowire interpolation task are presented below. The task uses a 80-10-10 randomized split over all structures in the dataset. Figure 9 depicts the distributions of DOS, cDOS, and injection velocity values across all energy bins and fermi levels in the test, training, and validation sets. Units and scales are the same as presented in the error tables.

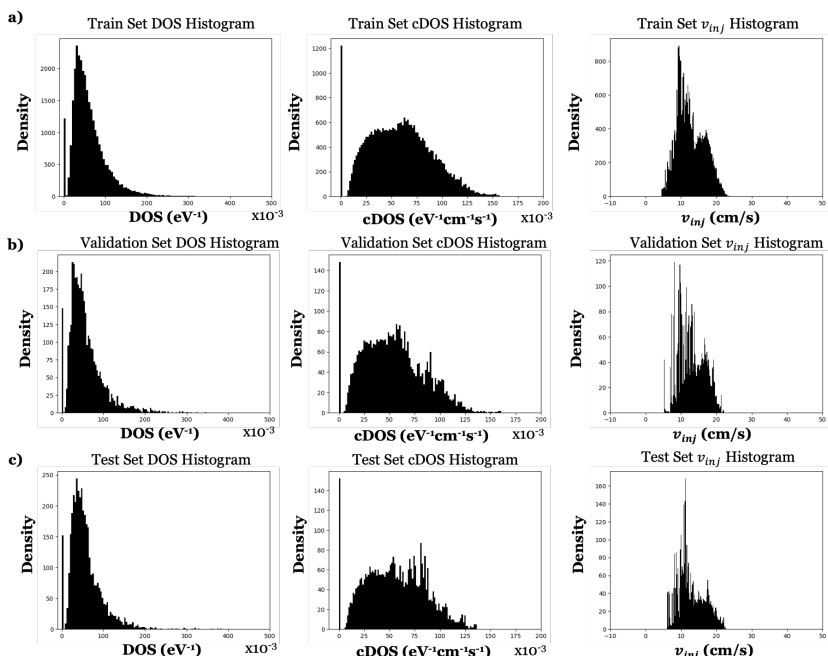

Figure 9: **Statistics of the In-Distribution Nanowire Experiment**: (a) Training Data Histograms for DOS, cDOS, and injection velocity. (b) Validation Data Histograms (c) Test Data Histograms. An 80/10/10 split is used over the entirety of the nanowire dataset. The training distribution qualitatively appears to be representative of the validation and test distributions

Table 8 includes the means and standard deviations across all energy bins and fermi levels of DOS, cDOS, and downstream derived quantities over the test, train, and validation datasets.

Table 8: **In-Distribution Nanowire statistics**: DOS ($eV^{-1}$), cDOS ($eV^{-1}cm^{-1}s^{-1}$), $N_{inv}$ ($cm^{-2}$), $I_{inv}$ (mA/$\mu$m), and injection velocity (cm/s) means over the train, validation, and test set for the out-of-distribution task are included below. Standard deviations are presented in parentheses below

| Set | Count | DOS [$\times 10^{-3}$] | cDOS [$\times 10^{-3}$] | $N_{inv}$ [$\times 10^{10}$] | $I_{inv}$ [$\times 10^{-3}$] | $v_{inj}$ [$\times 10^{6}$] |
|---|---|---|---|---|---|---|
| Train Stats | 305 | 61.40 (43.89) | 58.28 (31.05) | 1.23 (2.07) | 36.33 (53.86) | 12.65 (3.85) |
| Validation Stats | 37 | 58.10 (41.91) | 54.01 (29.90) | 0.96 (1.55) | 32.73 (48.23) | 12.89 (3.63) |
| Test Stats | 38 | 57.67 (39.49) | 56.23 (29.49) | 1.14 (1.91) | 35.06 (51.92) | 12.86 (3.73) |

Table 9: Hyperparameter specifications of EBFormer

| Hyperparameters | In-Distribution Nanosheet | | OOD Nanosheet & Nanowire | |
|---|---|---|---|---|
| | EBFormer-l0 | EBFormer-l1 | EBFormer-l0 | EBFormer-l1 |
| Message Passing Layers | 4 | 4 | 4 | 4 |
| Boundary Attention Heads | 1 | 1 | 1 | 1 |
| $\mathbf{w}_{\text{DOS}}/\mathbf{w}_{\text{cDOS}}$ | 3 | 3 | 3 | 3 |
| Cutoff Radius (Å) | 5 | 5 | 5 | 5 |
| Hidden Dimension | 400 | 215 | 122 | 64 |
| Learning Rate | 0.005 | 0.005 | 0.005 | 0.005 |
| Batch Size | 5 | 5 | 5 | 5 |

## D  IMPLEMENTATION DETAILS

We compare four architectures over a variety of configurations on the in-distribution and out-of-distribution tasks. EBFormer is characterized over a variety of training modes, equivariance degrees, and loss functions. The hyperparameters used for the invariant and l=1 equivariant approaches are included in Table 9. As a baseline, we include an MLP with a similar configuration as DOSTransformer Lee et al. (2023), which uses initial atomic embeddings as input to derive DOS and cDOS using a joint readout. The architecture has three hidden layers before the two readout heads, with hidden dimensions of 1000 for the interpolation task, and 300 for extrapolation to maintain weight parity.

Our GNNs are based off NequIP equivariant architectures Batzner et al. (2022). The ablation implementations are identical to the embedding layers in EBFormer, with the same shared hidden dimension, layer count, and cutoff as presented in Table 9. The invariant weight parity models use hidden dimension 1100 and 122 for the in-distribution and out-of-distribution tasks, respectively. The equivariant weight-parity models use hidden dimension 630 and 201 for the in-distribution and out-of-distribution tasks.

DOSTransformer hyperparameters are selected from the default configuration released with the code Lee et al. (2023). The readout layer is branched to two readout heads to facilitate joint training to fairly compare DOSTransformer with EBFormer, as we noted joint training led to a significant improvement in performance for our model. Hidden dimension is 256 for the in-distribution task, and 80 for the out-of-distribution task.

All models optimize over the same L1-loss functions $\mathcal{L}_{L/G}$ defined in Section 3.1 with $\mathbf{w}_{\text{DOS}}/\mathbf{w}_{\text{cDOS}} = 3.0$. We attempt to maintain weight parity across compared networks to avoid confounding effects of scaling Qu & Krishnapriyan (2024), which we accomplish by changing model hidden dimensions. This has the effect of scaling most of the main weight matrices involved in the network, including attention and convolution mechanisms. All the neural network code is written in PyTorch-CUDA Paszke et al. (2019). Training and inference were performed on an NVIDIA A100 GPU. Result tables with MAE and RMSE losses include mean (and standard deviation in parentheses below) over three runs of each model, each with a different seed. Seeds for NequIP-based approaches and EBFormer are selected from {123, 246, 369} for the model parameters, and {456, 912, 368} for dataset-related randomization, including data shuffling. The MLP and DOSTransformer use seeds {0, 1, 2}. Models on the result tables were selected based on minimum validation loss over 500 epochs of model training.

### D.1  VISUALIZATION OF BOUNDARY EMBEDDINGS

Figure 10 includes a visualization of molecular, bulk crystal, and mesoscale nanostructure systems, the last of which is the focus of EBFormer. We note that in the first two cases, local message-passing approaches are sufficient to capture the entire system geometry (notice the relatively few atoms required to define the entire system). However, mesoscale effects such as quantum confinement and crystal cleaving orientation lead to variation in material properties which require global augmentation. When applied to materials without boundary planes (such as molecular systems or

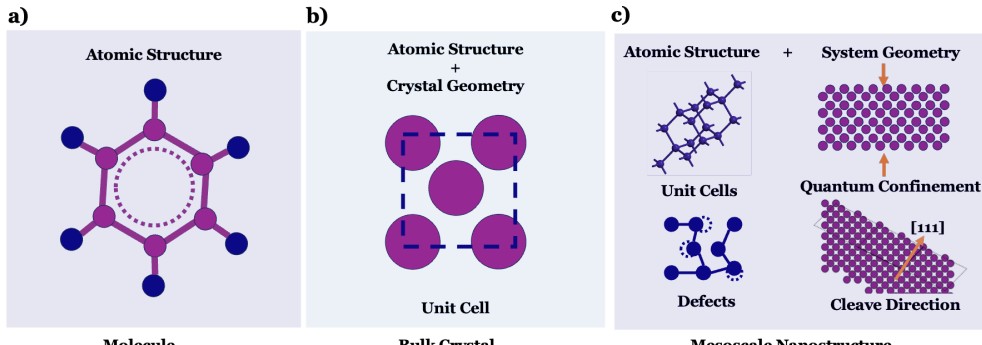

Figure 10: Visualization of (a) atomic (b) bulk crystal and (c) mesoscale nanostructure systems, which EBFormer focuses on

bulk crystals), our architecture reduces to the GNN we use for local convolution – in our case, a modified version of NequIP.

Figure 11 shows a visualization of the boundary embeddings for the nanowire and nanosheet structures. For the nanosheet, there are two boundaries; one at the top surface and one at the bottom of the nanosheet, in this case between the silicon and vacuum. In a full SOI gate-stack, this interface could be between the silicon and the bottom and gate oxides. For the nanowire, we have four boundary embeddings; one for each face of the nanostructure, again between the silicon and vacuum. Each boundary embedding interacts with all atoms in the boundary to atom cross-attention, while each atom collects information from both or all four of the boundaries for the nanosheet or the nanowire case, respectively. The interaction mechanisms are described in the main text in Equation 6.

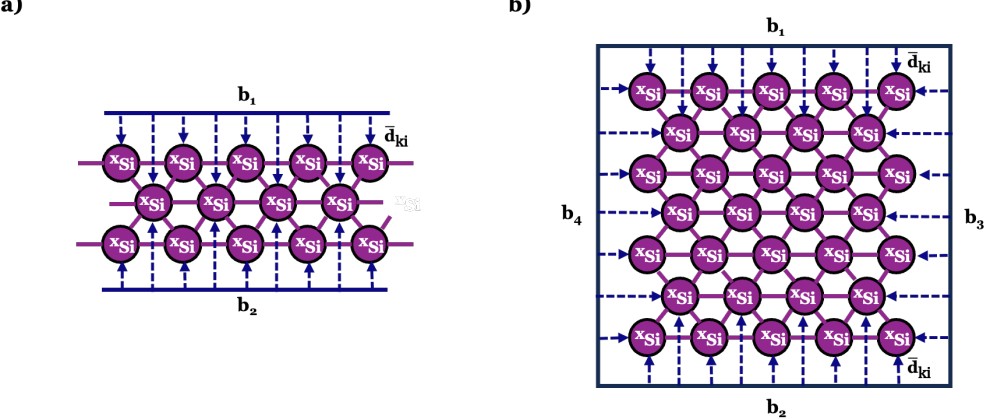

Figure 11: Visualization of the position and count of the boundary embeddings for the nanosheet (a) and nanowire (b) structures

## D.2 SENSITIVITY ANALYSIS

We conduct a sensitivity analysis on the relative weights of cDOS and DOS in the loss function, and also train separate models on each quantity individually, combining their predictions for injection velocity. We see that joint training yields a substantial improvement in injection velocity predictions, reinforcing the importance of physical consistency between DOS and cDOS for accurate downstream performance. Moreover, EBFormer maintains stable accuracy across a wide range of weight ratios, demonstrating robustness to a range of hyperparameter choices.

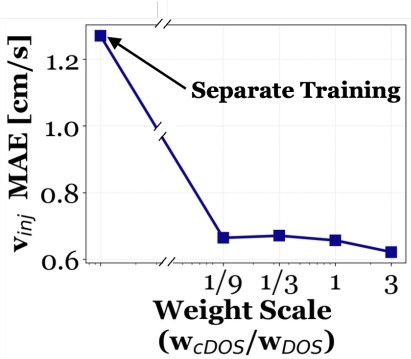

Figure 12: Sensitivity analysis of relative weighting of DOS and cDOS for the in-distribution nanosheet experiment for EBFormer

# E  QUALITATIVE VISUALIZATION

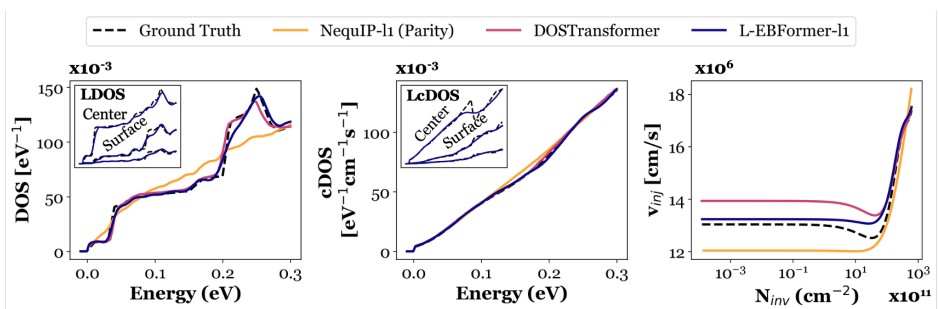

Figure 13: **Interpolation Qualitative Analysis** on a structure from the test set, 35 atomic layers of silicon cleaved along $\langle 100 \rangle$

Figure 13 compares model predictions on DOS, cDOS, and injection velocity on a structure from the test set. The ground-truth DOS show distinctive step-like features characteristic in confined 2D materials Datta (2005), and depends on the film thickness. NequIP-l1 is unable to capture these features since it is unaware of global geometry. DOSTransformer and EBFormer show comparable performances on DOS/cDOS, with EBFormer demonstrating improved injection velocity accuracy. The inset panel also shows high-fidelity reproduction of local atomic properties by EBFormer. Further characterization of LDOS/cLDOS prediction errors can be found in Appendix Section H.

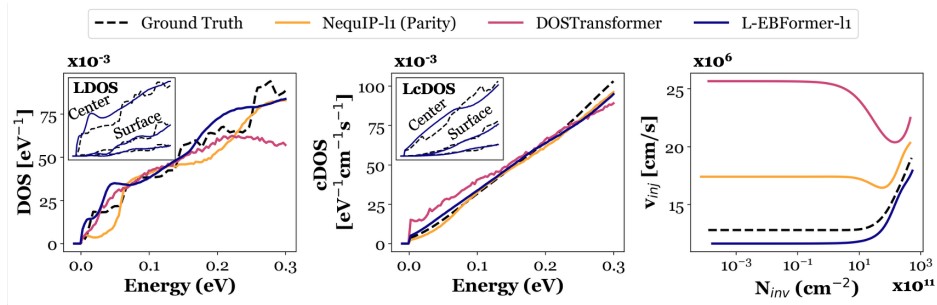

Figure 14: **Extrapolation Qualitative Analysis** on a structure from the test set with many more layers than in the training set, 77 atomic layers of silicon cleaved along $\langle 100 \rangle$

Extrapolation performance, visualized in Figure 14, qualitatively shows improved DOS predictions by EBFormer with approximate reproduction of confinement features in global and local DOS/cDOS. Note that this structure (at 77 layers) is significantly larger than the largest in the

training set (45 layers). We also observe significantly improved injection velocity prediction in extrapolation by EBFormer compared to the two other architectures.

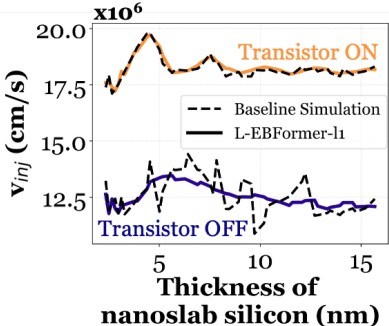

Figure 15: $\mathbf{v}_{inj}$ **vs Thickness**: EBFormer successfully learns the trend of injection velocity with nanosheet thickness

Finally, Figure 15 shows the variation of On and Off injection velocities for nanosheets (defined by thresholding inversion charge Rahman et al. (2003)). We see that EBFormer successfully reproduces the trend of currents with variation in thickness, an important task for transistor design that currently has garnered research interest Park et al. (2025).

# F  ERROR ANALYSIS OF $\mathrm{v}_{inj}$

Table 10: Error statistics and DOS–$J_x$ correlation quality. Units are as in the main text, with best values in bold.

| Model | Params [M] | DOS [$\times 10^{-3}$] | | cDOS [$\times 10^{-3}$] | | $N_{inv}$ [$\times 10^{11}$] | | $I_{inv}$ [$\times 10^{-3}$] | | $v_{inj}$ [$\times 10^{6}$] | | $R^2_{N_{inv}-I_{inv}}$ |
|---|---|---|---|---|---|---|---|---|---|---|---|---|
| | | MAE | RMSE | MAE | RMSE | MAE | RMSE | MAE | RMSE | MAE | RMSE | |
| MLP | 9.25 | 14.82 | 23.03 | 12.07 | 20.22 | 2.69 | 6.21 | 7.26 | 19.99 | 1.63 | 2.63 | -0.229 |
| | | (0.02) | (0.27) | (0.10) | (0.51) | (0.01) | (0.04) | (0.05) | (0.62) | (0.05) | (0.08) | (0.212) |
| NequIP-l0 | 1.39 | 5.59 | 9.14 | 2.40 | 4.51 | 0.45 | 1.35 | 1.43 | 4.26 | 0.87 | 1.97 | 0.677 |
| (Ablation) | | (0.01) | (0.07) | (0.04) | (0.05) | (0.00) | (0.03) | (0.02) | (0.08) | (0.00) | (0.16) | (0.039) |
| NequIP-l1 | 1.18 | 5.50 | 8.92 | 2.31 | 4.31 | 0.44 | **1.30** | 1.36 | 4.06 | 0.88 | 1.83 | 0.569 |
| (Ablation) | | (0.04) | (0.03) | (0.06) | (0.04) | (0.01) | (0.03) | (0.03) | (0.08) | (0.01) | (0.07) | (0.050) |
| NequIP-l0 | 9.98 | 5.53 | 9.17 | 2.29 | 4.39 | 0.44 | 1.36 | 1.37 | 4.18 | 0.91 | 2.24 | 0.632 |
| (Parity) | | (0.01) | (0.04) | (0.00) | (0.03) | (0.00) | (0.01) | (0.01) | (0.03) | (0.03) | (0.42) | (0.039) |
| NequIP-l1 | 9.46 | 5.46 | 8.91 | 2.30 | 4.27 | 0.44 | 1.31 | 1.36 | 4.02 | 0.85 | 1.70 | 0.602 |
| (Parity) | | (0.05) | (0.04) | (0.07) | (0.06) | (0.01) | (0.04) | (0.05) | (0.09) | (0.02) | (0.26) | (0.075) |
| DOSTransformer | 9.40 | **2.80** | 6.59 | 1.55 | 3.80 | 0.37 | 1.33 | 0.90 | 3.51 | 0.80 | 2.34 | 0.512 |
| | | (0.15) | (0.12) | (0.09) | (0.09) | (0.02) | (0.04) | (0.06) | (0.04) | (0.16) | (1.39) | (0.133) |
| EBFormer-l0 | 9.48 | 3.27 | 6.89 | 1.32 | 3.93 | 0.33 | 1.38 | 0.78 | 3.75 | 0.61 | 1.25 | 0.724 |
| | | (0.09) | (0.06) | (0.04) | (0.04) | (0.01) | (0.01) | (0.03) | (0.04) | (0.02) | (0.12) | (0.035) |
| EBFormer-l1 | 9.35 | 3.17 | 6.74 | **1.26** | 3.76 | 0.32 | 1.38 | 0.75 | 3.61 | 0.60 | 1.35 | 0.704 |
| | | (0.05) | (0.12) | (0.03) | (0.13) | (0.01) | (0.04) | (0.01) | (0.09) | (0.01) | (0.24) | (0.017) |
| L-EBFormer-l1 | 9.35 | 2.85 | **6.21** | **1.26** | **3.37** | **0.31** | **1.30** | **0.67** | **3.24** | **0.49** | **0.90** | **0.756** |
| | | (0.05) | (0.09) | (0.03) | (0.09) | (0.00) | (0.03) | (0.02) | (0.11) | (0.01) | (0.06) | (0.012) |

Injection velocity is a downstream function of inversion charge and current. These in turn are weighted integrals over DOS and cDOS in energy space, respectively, as defined in Section 2.1 in the main text. Errors propagate from the inversion charge and currents as follows:

$$v_{inj} = \frac{I_{inv}}{N_{inv}} \implies \frac{\Delta v_{inj}}{v_{inj}} = \frac{\Delta I_{inv}}{I_{inv}} - \frac{\Delta N_{inv}}{N_{inv}}$$

This implies that low injection velocity error requires (1) accurate $N_{inv}$ and $I_{inv}$ predictions and (2) similar magnitudes of $N_{inv}$ and $I_{inv}$ relative errors. Consistent relative errors in the inversion charge and current imply that these errors fall along the y=x line when plotted against each other. We

quantify this error by calculating the coefficient of determination of the relative errors in $N_{inv}$ and $I_{inv}$ using the y=x line as the best fit ($R^2_{N_{inv}-I_{inv}}$). We calculate the errors over the in-distribution nanosheet dataset and present the results in Table 10.

We see a strong correlation between the closeness of fit as given by the coefficient of determination and the accuracy in injection velocity, and note that EBFormer achieves the highest degree of covariance of $N_{inv}$ and $I_{inv}$. These quantities are derived from DOS and cDOS, which implies that these neural-network predictions also have a high covariance. Therefore, this shows that EBFormer's high success in injection velocity performance is due to accurate predictions as well as having learned the physical covariance of DOS and cDOS better than the benchmarked models.

## G  Timing and Memory Consumption

Figure 4 in the main text compares runtimes over a variety of structures for the empirical tight-binding method and EBFormer. Tight-binding is timed on an Intel Xenon Gold 6330 over 16-64 parallel processes. EBFormer inference is timed on an NVIDIA A100 GPU. The prediction time of the neural network only includes the inference time, and excluded data movement and formation. We see that EBFormer achieves 4-5 orders of magnitude speed-up compared to the baseline simulation method. Furthermore, while tight-binding methods scale as $O(n^3)$ Li et al. (2023) where n is the number of atoms in a unit-cell, EBFormer scales linearly, permitting simulation of significantly larger systems.

### G.1  Timing and Memory Consumption

Timing and memory comparison and throughput are presented in Table 11 for the models in our study.

| Model | Parameters (M) | Inference Speed (Samples / s) | Inference Memory (Peak / Sample) MB | Training Speed (Samples / s) | Training Memory (Peak / Sample) GB |
|---|---|---|---|---|---|
| MLP | 9.25 | 502.2 | 53 | 77.7 | 0.16 |
| ablation-l0 | 1.39 | 89.7 | 25 | 26.8 | 0.14 |
| wp-l0 | 9.98 | 95.5 | 79 | 28.0 | 0.54 |
| ablation-l1 | 1.18 | 64.8 | 58 | 15.3 | 0.38 |
| wp-l1 | 9.46 | 64.6 | 176 | 18.0 | 1.68 |
| DOSTransformer | 9.40 | 124.0 | 55 | 27.1 | 0.20 |
| EBFormer-l0 | 9.48 | 30.5 | 126 | 8.3 | 3.89 |
| EBFormer-l1 | 9.35 | 16.4 | 109 | 4.3 | 3.84 |
| L-EBFormer-l1 | 9.35 | 18.3 | 109 | 4.4 | 3.84 |

Table 11: Model size, speed, and memory during inference and training.

As noted in our limitations, using SE(3) equivariant attention involving Clebsch-Gordon tensor products is generally an expensive operation Batzner et al. (2022); Geiger et al. (2022); Luo et al. (2024). This is reflected in the table above in terms of memory and time usage of NequIP and EBFormer, which are equivariant architectures built on e3nn. However, we note that there have been many works that implement equivariant message-passing with higher-order tensorial features in an efficient manner through bespoke engineering and limited order of equivariance Wang et al. (2024c); Batatia et al. (2024); Wang et al. (2024a). Furthermore, our application demonstrates equivariant features only up to $l = 1$, meaning we operate with only scalar and vector-type objects. Interactions of these quantities require implementation only of vector-vector dot and cross products, scalar-vector multiplication, and scalar operations. While beyond the purview of our current work, replacing the complete machinery of e3nn and its generic treatment of irreps with a more tailor-made mechanism is a valuable and attainable next step.

Finally, we also observe that local training leads to minimal additional cost in memory or speed in both inference and training due to the fact that the intermediate values of LDOS/LcDOS are generated regardless of if they are used as a training signal. The improvement in downstream performance indicates that including these signals is an effective and low-cost way to improve performance given data availability.

## H  Local DOS/cDOS Errors

Tables 12 and 13 show the MAE and RMSE errors of LDOS and LcDOS along with all other model predictions. LDOS and LcDOS are the atom-wise predicted contributions to DOS/cDOS prior to

mean pooling that are used to generate the final mean DOS/cDOS. These quantities are only meaningful for architectures which make predictions on a per-atom basis – MLP and DOSTransformer do not generate local predictions, and therefore cannot generate or leverage the information in locally-projected quantities.

Table 12: **Nanosheet Interpolation Including LDOS/LcDOS Errors**: Interpolation accuracy including LDOS/LcDOS errors. Note that only L-EBFormer-l1 was trained on the local task. Models which cannot project predictions to a local basis (MLP and DOSTransformer) have no errors listed for the local tasks

| Model | Params [M] | DOS [×10⁻³] MAE | RMSE | LDOS [×10⁻³] MAE | RMSE | cDOS [×10⁻³] MAE | RMSE | LcDOS [×10⁻³] MAE | RMSE | $N_{inv}$ [×10¹¹] MAE | RMSE | $I_{inv}$ [×10⁻³] MAE | RMSE | $v_{inj}$ [×10⁶] MAE | RMSE |
|---|---|---|---|---|---|---|---|---|---|---|---|---|---|---|---|
| MLP | 9.25 | 14.82 (0.02) | 23.03 (0.27) | – | – | 12.07 (0.10) | 20.22 (0.51) | – | – | 2.69 (0.05) | 6.21 (0.62) | 7.26 (0.05) | 19.99 (0.62) | 1.63 (0.05) | 2.63 (0.08) |
| NequIP-l0 (Ablation) | 1.39 | 5.59 (0.01) | 9.14 (0.08) | 1055.06 (9.03) | 1710.27 (5.08) | 2.40 (0.05) | 4.51 (0.06) | 1058.15 (27.16) | 1754.87 (21.45) | 0.45 (0.00) | 1.35 (0.03) | 1.43 (0.03) | 4.26 (0.10) | 0.87 (0.01) | 1.97 (0.19) |
| NequIP-l1 (Ablation) | 1.18 | 5.50 (0.05) | 8.92 (0.04) | 1029.29 (49.49) | 1741.76 (74.85) | 2.31 (0.06) | 4.31 (0.11) | 974.32 (51.10) | 1773.18 (92.41) | 0.44 (0.01) | **1.30** (0.04) | 1.36 (0.04) | 4.06 (0.10) | 0.88 (0.02) | 1.83 (0.08) |
| NequIP-l0 (Parity) | 9.98 | 5.53 (0.01) | 9.17 (0.05) | 993.23 (14.01) | 1600.48 (24.00) | 2.29 (0.00) | 4.39 (0.04) | 985.88 (22.61) | 1625.68 (15.52) | 0.44 (0.00) | 1.36 (0.02) | 1.37 (0.01) | 4.18 (0.03) | 0.91 (0.04) | 2.24 (0.50) |
| NequIP-l1 (Parity) | 9.46 | 5.46 (0.06) | 8.91 (0.06) | 981.23 (35.68) | 1693.83 (70.94) | 2.30 (0.07) | 4.27 (0.07) | 947.03 (34.15) | 1779.91 (81.19) | 0.44 (0.01) | 1.31 (0.05) | 1.36 (0.06) | 4.02 (0.10) | 0.85 (0.02) | 1.70 (0.31) |
| DOSTransformer | 9.40 | **2.80** (0.15) | 6.59 (0.12) | – | – | 1.55 (0.09) | 3.80 (0.09) | – | – | 0.37 (0.02) | 1.33 (0.04) | 0.90 (0.06) | 3.51 (0.04) | 0.80 (0.16) | 2.34 (1.39) |
| EBFormer-l0 | 9.48 | 3.27 (0.10) | 6.89 (0.07) | 1005.10 (10.87) | 2101.54 (62.78) | 1.32 (0.05) | 3.93 (0.04) | 853.81 (34.00) | 1880.41 (94.95) | 0.33 (0.01) | 1.38 (0.01) | 0.78 (0.03) | 3.75 (0.05) | 0.61 (0.02) | 1.25 (0.14) |
| EBFormer-l1 | 9.35 | 3.17 (0.06) | 6.74 (0.14) | 1014.96 (14.18) | 2108.01 (53.59) | **1.26** (0.03) | 3.76 (0.16) | 819.26 (11.23) | 1703.02 (21.65) | 0.32 (0.01) | 1.38 (0.05) | 0.75 (0.02) | 3.61 (0.11) | 0.60 (0.01) | 1.35 (0.28) |
| L-EBFormer-l1 | 9.35 | 2.85 (0.06) | **6.21** (0.10) | 3.19 (0.06) | 7.67 (0.14) | **1.26** (0.04) | **3.37** (0.11) | 1.99 (0.04) | 4.92 (0.12) | **0.31** (0.00) | **1.30** (0.03) | 0.67 (0.02) | **3.24** (0.13) | **0.49** (0.01) | **0.90** (0.07) |

Table 13: **Nanosheet Extrapolation Including LDOS/LcDOS Errors**: Extrapolation accuracy including LDOS/LcDOS errors. Note that only L-EBFormer-l1 was trained on the local task. Models which cannot project predictions to a local basis (MLP and DOSTransformer) have no errors listed for the local tasks

| Model | Params [M] | DOS [×10⁻³] MAE | RMSE | LDOS [×10⁻³] MAE | RMSE | cDOS [×10⁻³] MAE | RMSE | LcDOS [×10⁻³] MAE | RMSE | $N_{inv}$ [×10¹¹] MAE | RMSE | $I_{inv}$ [×10⁻³] MAE | RMSE | $v_{inj}$ [×10⁶] MAE | RMSE |
|---|---|---|---|---|---|---|---|---|---|---|---|---|---|---|---|
| MLP | 0.99 | 27.69 (5.88) | 41.18 (9.80) | – | – | 23.88 (1.56) | 37.09 (1.72) | – | – | 2.38 (0.28) | 5.15 (0.20) | 8.39 (0.43) | 17.93 (1.72) | 7.94 (0.12) | 13.20 (1.35) |
| NequIP-l1 (Parity) | 1.04 | 8.82 (0.30) | 11.70 (0.33) | 975.21 (141.96) | 1574.29 (268.93) | 4.39 (0.04) | 6.05 (0.06) | 913.48 (161.69) | 1511.96 (336.40) | 1.22 (0.06) | 2.44 (0.06) | 2.68 (0.06) | 5.64 (0.34) | 2.37 (0.09) | 2.98 (0.10) |
| DOSTransformer | 0.95 | 11.88 (1.18) | 16.89 (0.79) | – | – | 7.11 (0.63) | 10.16 (0.50) | – | – | 1.42 (0.20) | 3.21 (0.28) | 4.20 (0.92) | 8.57 (1.66) | 4.99 (2.31) | 7.34 (3.61) |
| EBFormer-l0 | 0.94 | 9.12 (0.19) | 12.42 (0.17) | 940.02 (22.33) | 1694.37 (34.84) | 4.37 (0.26) | 7.41 (0.23) | 797.94 (33.99) | 1220.09 (24.82) | 1.35 (0.17) | 3.04 (0.21) | 2.42 (0.16) | 6.61 (0.17) | 1.00 (0.07) | 1.32 (0.05) |
| EBFormer-l1 | 0.91 | 7.97 (0.23) | 10.89 (0.25) | 831.37 (17.69) | 1497.16 (121.50) | **3.37** (0.23) | **5.39** (0.26) | 721.85 (113.18) | 1043.88 (181.63) | **1.00** (0.09) | **2.29** (0.10) | 1.81 (0.16) | 4.69 (0.25) | 0.89 (0.09) | **1.24** (0.05) |
| L-EBFormer-l1 | 0.91 | **7.52** (0.24) | **10.55** (0.44) | 9.49 (0.23) | 14.53 (0.74) | 4.58 (0.29) | 6.98 (0.62) | 7.01 (0.41) | 10.85 (0.57) | 1.03 (0.02) | 2.48 (0.15) | 2.76 (0.35) | 6.60 (0.87) | 0.97 (0.08) | 1.35 (0.06) |

Atom-local quantities such as LDOS and LcDOS can vary significantly over nanostructures. Furthermore, capturing this spatial variation of material properties can have significant impact on downstream device properties Jiang et al. (2008). For this reason, EBFormer is designed to be able to predict atom-local quantities. Table 12 demonstrates the high accuracy of L-EBFormer-l1 on the inference task of LDOS/LcDOS. In addition, we see that training on local quantities leads to *improvement* in performance in in-distribution regimes. This indicates that information regarding spatial variation of physical quantities is helpful to accurately reproduce physical quantities in systems with structure.

Note that out of all the models presented in the tables, only L-EBFormer-l1 was trained to optimize local targets. All other networks were trained to minimize error on the DOS/cDOS after mean-pooling. Therefore, the graph networks presented in these tables have high losses.

## I  TRANSFER LEARNING

The out-of-distribution task tested the zero-shot ability of EBFormer to make predictions on systems with confined dimensions larger than those in the training set. This was accomplished by training the model on 15-45 atomic layers of silicon of all cleaving orientations and strain, and testing on silicon

nanosheets of 46-100 atomic layers. Simulation of systems with large number of atoms rapidly becomes computationally infeasible, which makes extrapolation performance an important metric to quantify.

Table 14: **Extrapolation and Transfer Learning Errors** DOS ($eV^{-1}$), cDOS ($eV^{-1}cm^{-1}s^{-1}$), $N_{inv}$ ($cm^{-2}$), $I_{inv}$ ($mA/\mu m$), and injection velocity (cm/s) errors on the nanosheet extrapolation task are show below. In addition, performance on the same silicon test-task of 46-100 atomic layers is shown on models trained with 15-100 atomic layers of germanium added to the 15-45 atomic-layer silicon dataset

| Model | Params [M] | DOS [$\times 10^{-3}$] | | cDOS [$\times 10^{-3}$] | | $N_{inv}$ [$\times 10^{11}$] | | $I_{inv}$ [$\times 10^{-3}$] | | $v_{inj}$ [$\times 10^{6}$] | |
|---|---|---|---|---|---|---|---|---|---|---|---|
| | | MAE | RMSE | MAE | RMSE | MAE | RMSE | MAE | RMSE | MAE | RMSE |
| EBFormer-l0 | 0.94 | 9.12 | 12.42 | 4.37 | 7.41 | 1.35 | 3.04 | 2.42 | 6.61 | 1.00 | 1.32 |
| | | (0.19) | (0.17) | (0.26) | (0.23) | (0.17) | (0.17) | (0.16) | (0.21) | (0.17) | (0.14) |
| EBFormer-l1 | 0.91 | 7.97 | 10.89 | **3.37** | **5.39** | 1.00 | 2.29 | **1.81** | **4.69** | 0.89 | 1.24 |
| | | (0.23) | (0.25) | (0.23) | (0.26) | (0.09) | (0.10) | (0.25) | (0.25) | (0.09) | (0.05) |
| L-EBFormer-l1 | 0.91 | 7.52 | 10.55 | 4.58 | 6.98 | 1.03 | 2.48 | 2.76 | 6.60 | 0.97 | 1.35 |
| | | (0.24) | (0.44) | (0.29) | (0.62) | (0.02) | (0.15) | (0.35) | (0.87) | (0.08) | (0.06) |
| **Ge+**EBFormer-l0 | 0.94 | 8.24 | 11.55 | 3.91 | 6.89 | 1.13 | 2.74 | 2.20 | 6.22 | 0.77 | 1.09 |
| **Ge+**EBFormer-l1 | 0.91 | 8.02 | 10.98 | 3.96 | 5.91 | 1.18 | 2.54 | 2.22 | 5.28 | 0.76 | 1.17 |
| **Ge+**L-EBFormer-l1 | 0.91 | **6.51** | **9.35** | 3.44 | 5.53 | **0.80** | **2.21** | 1.86 | 4.88 | **0.71** | **1.07** |

However, as the paradigm of foundation models becomes increasingly central to materials informatics Batatia et al. (2024); Kovács et al. (2025), understanding the ability of our model to transfer information across chemical systems is also of interest. It is conceivable that a researcher exploring the properties of a novel material systems such as III-V semiconductor nanosheets may use a pretrained model with knowledge of confinement physics over a large range of nanostructures of traditional materials, and finetune with a few, small, cheaply generated examples of their new system. They may then desire to use the model to infer the behavior of nanostructures of various geometries comprised of the novel material.

We conduct an additional experiment to study EBFormer's potential in this setting by quantifying its ability to transfer geometric knowledge across chemical species. We achieve this by augmenting the out-of-distribution training dataset with germanium nanosheets of all cleaving orientations and strains with thickness between 15-100 atomic layers. This has the effect of including geometric information of larger (germanium) nanoslabs to the test set of small (silicon) nanoslabs. We then test performance on the same test set of large silicon nanoslabs as studied in the out-of-distribution evaluation in Section 4.2, which becomes in-distribution in terms of chemical composition and geometry individually, but not both simultaneously. The results are appended to the losses in the extrapolation task, with models trained with germanium labeled with the prefix **Ge+**. Losses are summarized in Table 14.

We notice minor improvements or comparable performance between EBFormer-l1 and the same model when augmented with germanium information. However, we see significant improvement in performance for the L-EBFormer model, indicating that EBFormer is able to leverage geometric information using spatial variation of atom-local quantities to improve global and local predictions.

## J PROOF OF EQUIVARIANCE

In this section, we prove that EBFormer is $SO(3)$ Equivariant. This implies that the updated embeddings of atomic and boundary nodes $\left(\mathbf{x_i}^{(t+1)}, \mathbf{b_k}^{(t+1)}\right)$ remain equivariant after the message-passing and update phases. Formally, for any orthogonal matrix $Q \in \mathbb{R}^{n \times n}$, we need to prove that EBFormer satisfies

$$Q\mathbf{x_i}^{(t+1)}, Q\mathbf{b_k}^{(t+1)} = \text{EBFormer}\left(Q\mathbf{x_i}^{(t)}, Q\mathbf{b_k}^{(t)}\right) \tag{11}$$

EBFormer contains two stages: (i) The local equivariant convolution, and (ii) The global equivariant attention between atom and the boundary nodes. If each stage is equivariant, then their composition is equivariant. The equivariance of the local convolution follows from prior work Batzner et al.

(2022); Satorras et al. (2021). We therefore focus on the cross-attention between boundary and atom nodes, specifically Eqs. (4) and (5) in the main text. If these two equations are equivariant, then the entire EBFormer is equivariant.

## J.1 ATOM NODES

Assume $\mathbf{x}_i^{(t)}$ and $\mathbf{b}_k^{(t)}$ transform equivariantly under $SO(3)$. We show that $\mathbf{x}_i^{(t+1)}$ is also equivariant:

$$\mathbf{x}_i^{(t+1)} = \mathbf{x}_i^{(t)} + \sum_k \alpha_{ik} \left( \mathbf{b}_k^{(t)} \overset{\mathbf{W}_V^A}{\otimes} Y_J^{(l)}(\vec{\mathbf{d}}_{ik}) \right) \tag{12}$$

Here, $Y_J^{(\ell)}(\vec{\mathbf{d}}_{ik})$ denotes the spherical-harmonic expansion of the relative vector from atom $i$ to boundary plane $k$, which is equivariant by construction. The tensor product with $\mathbf{b}_k^{(t)}$ uses Clebsch–Gordan coefficients Thomas et al. (2018), thus preserving equivariance (similar to standard equivariant graph convolutions). The attention weights $\alpha_{ik}$ are *scalars* computed from queries $\mathbf{q}_k$ and keys $\mathbf{k}_i$ via equivariant linear maps applied independently to each irrep, followed by tensor products and extraction of scalar (invariant) components. Hence $\alpha_{ik}$ are invariant under rotations. Multiplying invariant scalars with equivariant features preserves equivariance, and summing over $k$ messages preserves equivariance by closure under addition. Therefore, $\mathbf{x}_i^{(t+1)}$ is equivariant given $\mathbf{x}_i^{(t)}$ and $\mathbf{b}_k^{(t)}$ are equivariant.

## J.2 BOUNDARY NODES

Similarly, assume $\mathbf{x}_i^{(t)}$ and $\mathbf{b}_k^{(t)}$ are $SO(3)$-equivariant. We show $\mathbf{b}_k^{(t+1)}$ is equivariant:

$$\mathbf{b}_k^{(t+1)} = \mathbf{b}_k^{(t)} + \sum_i \alpha_{ki} \left( \text{EquivLinear}_V^B \left( \mathbf{x}_i^{(t)} \middle| \text{MLP}_V^B \left( \mathbf{f}_{ik} \right) \right) \right) \tag{13}$$

The features $\mathbf{f}_{ik}$ are invariant, as they are functions of scalar distance quantities. The map $\text{EquivLinear}_V^B$ acts as an irrep-wise linear operator and therefore preserves equivariance. The attention weights $\alpha_{ki}$ are scalar invariants constructed as in the atom node case. Products of invariant scalars with equivariant features are equivariant, and summation over $i$ preserves equivariance. Hence $\mathbf{b}_k^{(t+1)}$ is equivariant given $\mathbf{x}_i^{(t)}$ and $\mathbf{b}_k^{(t)}$ are equivariant.

