# OpenReview forum: "Geometry-Aware Equivariant Attention for Scalable Nanoelectronic Property Prediction"
_ICLR.cc/2026/Conference — Submitted to ICLR 2026_

### Official Review · Reviewer_fHnK · 2025-10-18

**Soundness:** 3
**Presentation:** 3
**Contribution:** 3
**Rating:** 4
**Confidence:** 3

**Summary:**

The primary methodological contribution of the paper is a symmetry-aware message passing scheme between atoms and boundary nodes of finite nanostructures. The work was motivated with the observation that fully-connected attention mechanisms training on Si nanostructures learnt two-fold interactions, one which covered local neighborhood interactions, and another which connected atoms with the boundary. By developing a boundary attention mechanism, the authors include this learned effect as an inductive bias in the model, thus increasing expressivity while avoid the unfavorable scaling which would otherwise result from fully-connected attention mechanisms.

Overall, I think it's an interesting approach to model properties heavily affected by quantum confinement at the mesoscale. However I have a few concerns and questions detailed below.

**Strengths:**

I appreciate the depth the authors went into to generate custom domain-specific dataset, and curate and release it. The overall quality of the work is high and the authors are thorough in gathering results and ablation studies.

**Weaknesses:**

I'm not sure if this work is suitable for a general ML audience.

The paper is a bit cryptically written, and it is quite difficult to understand the problem that the author are trying to treat. The explanation of quantum confinement effects in nanostructures is understandable, but I had to read halfway through the paper before I understood that the authors designed a tool for DOS prediction in nanostructures which contain both local periodicity and finite overall size. Without some background in device modeling, I don't know if I would have understood the relevance of the problem, and this makes me think it might be better suited to a domain-specific journal.

I'm also not sure if this method can be applied broadly. I can see generalization ability to other mesoscale crystalline structures, but I'm not sure where the results could apply beyond that for the ML materials community.

Finally, looking at Figures 12-14, the results seem to be not yet great. I assume that the goal of this model would be to instantaneously regenerate the DOS/cDOS and re-evaluate the injection velocity upon changes in the atomic structures (such as adding strain, changing the cleavage plane, decreasing the nanosheet thickness, etc) providing a tool for computational design of silicon nanosheets in transistor channels. However I think these errors might be higher than the variations in DOS upon reasonable amounts of strain, for example, which challenges the practical utility.

I would make the paper less cryptic on what is being treated. A schematic illustrating the differences between molecular structures, periodic materials, and mesoscale nanostructures would be nice. I would also include Fig 5, and an example of a silicon nanowire structure in the main text.

I think some more details about the dataset should also be in the main text. Without looking at the appendix, I had no picture of what the model was being trained on.

**Questions:**

-Intuitively, I think there should be a trend in the effect of including the boundary nodes with respect to structure size. It seems however that the results were taken over all structures. Have the authors looked into how including the boundary attention mechanism changes performance when testing on structures with different sizes?

-When evaluating the boundary attention mechanism, the authors compare between a relatively small local cutoff (5A) and a fully-connected graph. Have they looked into instead using a slightly larger cutoff? Presumably, once the nanostructure is large enough, the properties at the 'center' converge to those of the bulk material, the question is only the exact size at which this occurs. I see that the attention learned from the fully-connected graph prioritizes local interactions and boundary connections. However, attention mechanisms are known to learn shortcuts  on the training data in scientific simulations, and maybe this is one of them. I wonder if simply using a cutoff of ~8 A would close the gap between the performance of the local and fully-connected networks.

---

> ### Author Response · Authors · 2025-12-03
>
> Thank you for your comments and detailed overview of our paper. We are happy to address the questions and concerns brought up:
>
> **Weakness 1:**
> > "The paper is a bit cryptically written, and it is quite difficult to understand the problem that the author are trying to treat."
>
> We apologize for our unclear presentation of the task EBFormer is designed to address. We have updated the main text and figures per your suggestions.
>
> **Weakness 2:**
> > "I'm also not sure if this method can be applied broadly"
>
> Our primary motivation for the development of EBFormer is to address the difficult technical challenge of characterizing the electronic properties of nanostructures. Transistors, image sensors, memories, solar panels, and LEDs are all examples of nanostructures which play a central role in defining our modern technological landscape, and whose properties depend on both structure and material composition. For example, SOI transistors with nanosheet silicon channels with 3nm thickness carry 20-30% more current than 6nm devices due to quantum confinement [1]. Atomic defects in LEDs lead to significant decrease in light emission [2]. TSMC and Samsung's upcoming 2nm nodes rely on nanowires similar to those in our response to reviewer LbYC [3]. In our work, silicon and germanium were chosen as case-studies due to their ubiquity and immediate, industry-critical applications, but EBFormer is applicable to various materials, geometries, and material phases.
>
> Since structural and atomic-scale material properties both impact final device properties, accurate simulations need to consider a large number of atoms that comprise the entire nanostructure and define its geometry. Standard physical simulations such as DFT or tight-binding might hours to days to solve realistic device geometries, while EBFormer requires seconds to minutes. An example pipeline integrating EBFormer is provided below:
>
> 1. **Atomic structure generation**: Determining the atomic structure of the nanodevice requires placing the various material structures in an approximate initial configuration, and using molecular dynamics to develop the interfacial bonds. Traditionally conducted with physics-based approaches such as DFT, tight-binding, or empirical force-fields, modern neural-network interatomic potentials (ex. NequIP) are now incorporated in industry-standard tools for nanoelectronic design [4]. This step captures interfacial strain, defects, and can be seeded with various initial atomic configurations for Monte-Carlo variability analyses.
> 2. **Microscopic electronic structure determination**: Given the atomic structure, the electronic structure of the system must be determined. This is a difficult, nonlocal quantum problem which can be computationally intractable. EBFormer is introduced to accelerate this step.
> 3. **Macroscopic electronic behavior determination**: Finally, microscopic electronic properties such as DOS and cDOS are translated to macroscopic device properties such as currents, light-capture efficiency, or luminosities using standard physical theory.
>
> [1] J. Ho Park et al. in IEEE EDL, Feb. 2025, doi: 10.1109/LED.2024.3507635.
>
> [2] I. Mártil, et al. J. Appl. Phys. 1 March 1997, doi: 10.1063/1.364294
>
> [3] Vaughan, O. Nat Electron 7, 1063 (2024)
>
> [4] S. Smidstrup, et al. J. Phys: Condens. Matter (APS), Vol. 32, pp. 015901 (2020)
>
> [5] K. Kaharudin et al 2020 J. Phys.: Conf. Ser. 1502 012045 DOI 10.1088/1742-6596

---

> ### Author Response · Authors · 2025-12-03
>
> **Weakness 3/Question 1:**
> > "However I think these errors might be higher than the variations in DOS upon reasonable amounts of strain, for example, which challenges the practical utility."
>
> As noted in the main text, architectural decisions between nodes typically yield 15-60% impact in quantities such as power, current, and electron mobility. Resolving the impact of these decisions requires at least <10% errors in predictions. DOSTransformer showed RMSE of 17% in-distribution and _52%_ in out-of-distribution inference of injection velocity (linearly proportional to transistor current), while EBFormer demonstrated errors of 6% in-distribution and _9%_ OOD, showing vastly improved performance compared to state-of-the-art architectures and presenting a path to practical applicability in device-technology co-optimization pipelines.
>
>
> **Question 2:**
> > "I wonder if simply using a cutoff of ~8 A would close the gap between the performance of the local and fully-connected networks."
>
> Simply increasing the model cutoff radius presents multiple problems, including the runaway growth of configuration space within the local environment (requiring similarly scaling amounts of training data), as well as the increased memory burden of construction and inference over large computational graphs. We also observe significant variation in ground-truth central atom LDOS/LcDOS between 35 and 77 atomic layers (corresponding to 29Å and 64Å, respectively) in Figures 12-13 in the Appendix, indicating that receptive fields of at least 32Å are necessary to capture the impact of the material boundaries on the central atom material properties. For reference, we found cutoff radii beyond ~10Å leads to out-of-memory errors over our dataset with batch size of 10 on an A100 GPU with 80 GB memory.

---

### Official Review · Reviewer_keTe · 2025-10-31

**Soundness:** 3
**Presentation:** 3
**Contribution:** 2
**Rating:** 2
**Confidence:** 4

**Summary:**

The author invents a method to embed the boundary/cell information and their interaction with the atoms' position into the network and uses this information to guide the prediction of the target properties. It has several benefits, such as computational efficiency, but some shortcomings also exist, which I will comment on later.

**Strengths:**

The author perform very good experiment include indistribution and out of distribution analysis. The presentation is good, the writing is very clear.

**Weaknesses:**

The method is not very novel; several similar methods that incorporate the boundary/cell information into the system exist. And this method has some intrinsic difficulties in transferring to a similar system.

**Questions:**

1. The box/boundary vector, as initialised, can be shifted by an arbitrary constant in the x/y/z direction without changing the shape. Is this method transferable to all these simulations?

2. How to deal with the supercell of the same system? Does the model, trained on small cells, automatically generalise to its supercell?

3. Can you compare the result with a randomly generated initial bound vector to ensure that the result improvement does not come from the overfitting?

---

> ### Author Response · Authors · 2025-12-03
>
> We thank the reviewer for their feedback and review, and address their questions as follows:
>
> **Weakness 1**:
> > " The method is not very novel; several similar methods that incorporate the boundary/cell information into the system exist."
>
> Our architecture is specifically designed to capture global geometric effects—such as boundary conditions, dimensional confinement, and cleaving orientations—that strongly influence the electronic properties of nanoscale devices that are challenging to handle by standard message-passing networks.
>
> The novelty of our approach lies in explicitly introducing boundaries into atomic graph networks, and introducing a framework to efficiently communicate requisite long-range information to and from atomic nodes via equivariant attention mechanisms to capture geometric confinement. Our design is motivated by an observation of sparsity in a full self-attention interaction over all atoms (highlighted in Appendix A "ARCHITECTURAL MOTIVATION"). When utilizing self-attention to capture global effects, the attention map exhibits an "N"-shaped pattern: a local diagonal component and two strong nonlocal “legs,” showing that every atom has significant interactions with atoms near the boundaries. Instead of performing costly all-to-all attention, we explicitly encode these interactions through boundary nodes, achieving the same effect while maintaining linear O(N) scaling, whereas most approaches incorporating long-range interactions scale worse [1, 2].
>
> This architecture is a physics-inspired extension that enables any local message-passing GNN to efficiently account for global geometric constraints arising from material interfaces. Beyond our work in the main text, we also demonstrate the applicability of our framework to the GAAFET architecture detailed in Table 3 in the main text. Further architectural motivation and discussion of novelty in included in Appendix A "ARCHITECTURAL MOTIVATION."
>
> [1] Giulia Luise et al. (2025). 10.48550/arXiv.2506.14665
>
> [2] Yusong Wang et al. (2024). Thirty-eighth Annual Conference on Neural Information Processing Systems.
>
> **Weakness 2/Question 1-3**:
> > " And this method has some intrinsic difficulties in transferring to a similar system. The box/boundary vector, as initialised, can be shifted by an arbitrary constant in the x/y/z direction without changing the shape. Is this method transferable to all these simulations? How to deal with the supercell of the same system? Does the model, trained on small cells, automatically generalise to its supercell? Can you compare the result with a randomly generated initial bound vector to ensure that the result improvement does not come from the overfitting?"
>
> EBFormer is designed to perform identically under all of the specified transformations.
>
> 1. **Shifting of the boundary vector/bounding box**: We note first that the boundary embedding represents an interfacial plane. An atom/boundary vector represent the unique shortest vector that is normal to the boundary plane connecting the atom to the plane's surface. The boundary-atom interaction mechanism and the local network (and therefore the entirety of EBFormer) are thus invariant to transformations of the bounding box perpendicular to the plane normal vector. Along the normal vector, the boundary plane is chosen to be a constant distance from the extremal atomic position projected along said vector. Since the boundary plane transforms with the input atomic structure independent of x/y/z coordinate choice, EBFormer is therefore invariant to coordinate transformations and shifts of the bounding box.
> 2. **Supercells of the same system**: EBFormer generalizes to supercells of the system automatically. This is due to the fact that periodic images are necessarily replicas perpendicular to the boundary normal vector. As noted in the previous point, the boundary-atom vector is therefore identical. We consider an inductive argument: the initial atomic and boundary embeddings are identical for a unit and supercell per initialization. The boundary-to-atom attention is also identical across both cases, as the attention mechanism (Eq. 6) is identical over replicated rows of the key/value matrices. This therefore implies that the atom-to-boundary attention is also identical (Eq. 7), since the atom and updated boundary embeddings are both identical. Finally, the local atomic updates are also identical due to the periodic invariance of the graph network [1]. Therefore, each round of message-passing and boundary/atom cross-attention are identical on supercells of the input structure, and therefore the final output is also invariant, leading to automatic generalizability.
>
> **The identified limitations are therefore not applicable to our approach.** Maintance of these symmetries was our primary motivation for the introduction of boundaries as planar interfaces.
>
> [1] Yan, K., et al. (2022). (NeurIPS 2022).

---

### Official Review · Reviewer_8WJe · 2025-10-31

**Soundness:** 3
**Presentation:** 3
**Contribution:** 3
**Rating:** 6
**Confidence:** 3

**Summary:**

This paper introduces EBFormer, a geometry-aware equivariant neural network designed to predict electronic properties of nanostructures (nanosheets and nanowires). The key innovation is augmenting local graph convolution (NequIP-style) with boundary cross-attention mechanisms to capture effects arising from global geometry. The authors demonstrate that explicitly encoding boundary interactions helps for accurate property prediction, particularly in out-of-distribution scenarios involving structures with dimensions beyond the training set.

**Strengths:**

1. **Well-motivated architecture**: The approach is physically grounded. The authors clearly motivate why boundary information is necessary for capturing quantum confinement effects in nanostructures, and the boundary cross-attention mechanism is an elegant solution that maintains O(N) scaling.

2. **Strong extrapolation performance**: The most impressive result is the model's ability to extrapolate to nanosheets with 46-100 atomic layers when trained only on 15-45 layers.

3. **Thorough experimental design**: The paper includes appropriate ablations (NequIP baselines, parameter-matched comparisons).

**Weaknesses:**

1. **Limited dataset scope**: The evaluation is restricted to Si and Ge nanosheets/nanowires with specific cleaving orientations. While the paper demonstrates interpolation and size extrapolation, the generalization to:
   - Other materials systems (III-V semiconductors, 2D materials, etc.)
   - Different boundary conditions
   - Novel nanostructure geometries
   remains unclear.

2. **Insufficient cross-geometry validation**: A critical validation experiment is missing: training on nanosheets and testing on nanowires (and vice versa). This would strongly demonstrate whether the model truly learns general boundary/confinement physics versus memorizing geometry-specific patterns. The current setup trains and tests on each geometry separately.

3. **Transfer learning underexplored**: While Appendix I shows some transfer learning from Ge to Si for thickness extrapolation, this is relatively limited. More systematic evaluation of:
   - Cross-material transfer (train on Si, test on Ge)
   - Cross-geometry transfer (nanosheets <-> nanowires)
   - Few-shot adaptation to new materials
   would significantly strengthen the claims about learning physical principles.

4. **Computational efficiency concerns**: Table 11 shows EBFormer is 3-6× slower than DOSTransformer and NequIP during inference, and training is even more impacted. While the authors acknowledge this and suggest future optimizations, this limits practical adoption for high-throughput screening.

**Questions:**

1. **Cross-geometry experiments**: Can you add experiments training on nanosheets and testing on nanowires? This would be very valuable for understanding whether boundary attention truly captures generalizable confinement physics.

2. **Boundary representation**: How sensitive is performance to the number and placement of boundary embeddings? What happens with more complex geometries (e.g., core-shell nanowires)?

3. **Comparison scope**: How would simpler approaches perform, such as:
   - Adding system-level features (thickness, aspect ratio) to a standard GNN?
   - Using a single global node instead of boundary-specific nodes?

---

> ### Author Response · Authors · 2025-12-03
>
> Thank you for your detailed and constructive feedback. We are encouraged that you found our work innovative and our architecture well-motivated.
>
> **Weakness 1**:
> > "While the paper demonstrates interpolation and size extrapolation, the generalization to: Other materials systems (III-V semiconductors, 2D materials, etc.), Different boundary conditions, Novel nanostructure geometries remains unclear."
>
> Due to the high computational cost of simulating our systems, we as of yet do not have data in our custom dataset for other material systems and properties. Generation of the dataset in its current form required ~4 months of computational wall time on three compute clusters with 112 processes hosted on Intel Xenon Gold 6330 processors. We do note that 2D material unit-cells do not require augmentation with boundary information due to the relatively small size of the unit cell, and therefore the absence of long-range unit cell geometric structure.
>
> **Weakness 2/Question 1**:
> > "Transfer learning underexplored: While Appendix I shows some transfer learning from Ge to Si for thickness extrapolation, this is relatively limited. More systematic evaluation of: Cross-material transfer (train on Si, test on Ge), Cross-geometry transfer (nanosheets <-> nanowires), Few-shot adaptation to new materials would significantly strengthen the claims about learning physical principles."
>
> Cross material transfer learning displays poor performance, mainly due to the fact that elemental embeddings are initialized to a random vector based on atomic species. Without any training data, novel materials lead to performance similar to random noise as the input vector is out of distribution for all learned weights. However, we have not validated our model on the remaining two tasks due to high training times, but will incorporate these experiments in future revisions of this work.
>
> **Question 3**:
> > "Comparison scope: How would simpler approaches perform, such as: Using a single global node instead of boundary-specific nodes?Adding system-level features (thickness, aspect ratio) to a standard GNN?""
>
> DOSTransformer was included as a comparison to EBFormer due to the collation of global information through the cross-attention mechanism between energy bins and local atomic embeddings. This in principle is similar to the inclusion of multiple global nodes which collect information across the entirety of the atomic graph per each energy bin, a relaxation of the inductive bias of including a single global node. However, we note that this approach cannot easily encode global long-range geometry information as the position of the global node in space must be made arbitrarily and independent of system geometry, meaning an approach such as DOSTransformer must omit spatial information when collecting global features. Inclusion of system-level features directly as vector input to the neural network is a simple means to incorporate global information, but has empirically shown poor extrapolation and out-of-distribution performance. However, a more thorough study is warranted as an extension of our manuscript in future revisions.
>
> **Weakness 4**:
> > "Table 11 shows EBFormer is 3-6× slower than DOSTransformer and NequIP during inference, and training is even more impacted. While the authors acknowledge this and suggest future optimizations, this limits practical adoption for high-throughput screening."
>
> While there is no standard metric for required accuracy, architectural decisions between nodes typically yield 15-60% impact in quantities such as power, current, and electron mobility. Resolving the impact of these decisions requires at least <10% errors in predictions. DOSTransformer showed RMSE of 17% in-distribution and _52%_ in out-of-distribution inference of injection velocity (linearly proportional to transistor current), while EBFormer demonstrated errors of 6% in-distribution and _9%_ OOD (Tables 1 and 2 in the main text). We also note that DOSTransformer is unable to predict local quantities such as LDOS and LcDOS due to global structure/energy cross-attention.
>
> Finally, we raise the point that EBFormer comfortably fits within a single A100 GPU, and provides 10$^4$-10$^5$ times speed-up compared to the baseline simulation while maintaining comparable accuracies; integration of EBFormer into existing simulation pipelines therefore provides significant speed-up without prohibitive cost. However, as the reviewer noted, future approaches to efficient equivariance such as those utilized by MACE/PAINN would provide a valuable extension to our work.

---

### Official Review · Reviewer_GH4c · 2025-10-31

**Soundness:** 1
**Presentation:** 3
**Contribution:** 1
**Rating:** 0
**Confidence:** 5

**Summary:**

The paper presents EBFormer, an SE(3)-equivariant architecture that augments local graph convolution with a boundary cross-attention mechanism to encode global geometry (e.g., confinement planes). The model is trained to predict DOS/cDOS and derived device metrics (injection velocity, ballistic current) for nanosheets and nanowires, using a custom, simulator-generated dataset (∼12k Si/Ge nanosheets; 380 Si nanowires) computed with tight-binding. EBFormer is compared against NequIP and DOSTransformer (plus an MLP), showing improvements in-distribution and on a thickness OOD split. Speedups vs. physics simulators are claimed (orders of magnitude).

**Strengths:**

Physically motivated design: boundary-aware equivariant attention is a plausible way to inject geometry. Clear target quantities: DOS/cDOS → injection velocity/current provide device-adjacent metrics with interpretable units. Some robustness checks: OOD across nanosheet thickness; ablations vs. local-only and parameter-parity settings. Computational efficiency: claimed orders-of-magnitude faster than physics simulators.

**Weaknesses:**

The paper’s evaluation is limited in both scope and rigor. It compares the proposed model only against NequIP, DOSTransformer, and a simple MLP baseline, omitting stronger and more diverse families such as MACE/MACE-MP, Allegro, PaiNN/NequIP-v2, GemNet-T (OC20 baselines), Matformer/COMFormer, and M3GNet. The label space is also narrow, focusing solely on DOS and cDOS while neglecting other physically significant quantities like band structures, effective masses, dielectric and optical responses, phonon spectra, elastic constants, scattering-aware transport, and device-level metrics that would better probe physical generalization. Moreover, the model’s external validity is questionable—no experiments are performed on established public datasets such as Matbench, JARVIS, or OC20/OC22, nor on experimental targets—making it difficult to assess general usefulness beyond the authors’ custom dataset. The claimed novelty of the boundary-attention mechanism is also uncertain, as similar global-context mechanisms could plausibly yield comparable effects, and the paper lacks comparative evidence to isolate what uniquely drives performance gains. Overall, since the work introduces both an in-house dataset and a “novel” model, it should be benchmarked on at least three to five standard datasets and compared with a wider range of five or more well-known models to substantiate its claims and demonstrate genuine methodological advancement.

**Questions:**

Public benchmarks: Can you report on Matbench (DOS/elastic), JARVIS, or OC20/OC22 tasks to test periodic/surface/generalization claims?

Particle count: What are the atom counts in each nanoparticle?

Cross-simulator validation: How does EBFormer trained on TB perform on a DFT-labeled subset (no simulator leakage)?

Broader chemistry: Results on III-V (GaAs, InP), wide-bandgap (GaN, SiC), oxides/heterostructures? Any transfer learning experiments?

Property breadth: Can you extend to band structures/effective masses, phonons/elastic moduli, dielectric spectra to stress different physical regimes?

Baselines: Why the models such as MACE/Allegro/PaiNN/GemNet/Matformer/COMFormer/M3GNet with parameter parity.

---

> ### Author Response · Authors · 2025-12-03
>
> Thank you for your feedback and review. We address your comments as follows:
>
> **Question 1/Weakness 2**
> > "... the model’s external validity is questionable—no experiments are performed on established public datasets such as Matbench, JARVIS, or OC20/OC22"
>
> EBFormer is designed to capture the effects of confinement due to boundaries or interfaces on material properties and their spatial variation, incorporating long-range confinement effects with local material variation in a scalable fashion. Capturing these effects is necessary to describe the properties of mesocale structures between tens to hundreds of atoms thick along their confined dimensions, which are ubiquitous in modern nanoelectronics. Simulation of such nanostructures requires a large unit-cell with many atoms to capture the geometry of the nanostructure, which is computationally-expensive. To the best of our knowledge, our dataset is the first publically-available to include these effects.
>
> EBFormer can in theory be applied to common baseline datasets including for bulk materials or molecules. However, the vast majority of the structures in these datasets do not include the material properties of confined or heterogeneous materials. This includes most of the Materials Project as well as OC20/OC22, which include energies and forces of adsorption and relaxation pathways of small molecules. **When applied to materials without boundary planes, our architecture reduces to the GNN we use for local convolution** -- in our case, a modified version of NequIP. Since this network has been extensively tested against a variety of standard material benchmarks, we did not prioritize testing on standard datasets.
>
> JARVIS-Heterostructure (and matbench_jdft2d in Matbench), MC2D, C2DB, and ML2DDB include exfoliation energies and other material properties of heterogenous material structures. However, they are limited to *exfoliated mono- and bilayers* of 2D materials, which again do not contain long-range confinement effects, rendering local GNN-approaches again sufficient. Finally, JARVIS-DFT includes 2D-bulk and 1D-bulk materials. However, these datasets again include heterstructures of Van der Waals-bonded lower-dimensional structures (i.e. monolayers and atomic chains), which do not require long-range information to capture confinement.
>
> **Weakness 1/Question 6**
> >"The paper’s ... compares the proposed model only against NequIP, DOSTransformer, and a simple MLP baseline, omitting stronger and more diverse families such as MACE/MACE-MP, Allegro, PaiNN/NequIP-v2, GemNet-T (OC20 baselines), Matformer/COMFormer, and M3GNet."
>
> We introduced EBFormer in this work as a methodology of incorporating long-range confinement information for prediction of local material properties in a scalable fashion. Capturing the effects of quantum confinement with GNN-approaches with finite cutoff radii is impossible as material interfaces leading to confinement extend beyond the receptive field of the model. Simply increasing the cutoff radius is also infeasible, as discussed in further detail in our response to Reviewer fHnK.
>
> Our work presents a generalizable framework to incorporate the effects of atomic interactions with boundary planes on local and global electronic structural variation. Specifically, we focus on structured materials with long-range order such as those dictating the electronic properties of modern nanoelectronic devices, which can be comprised of hundreds of thousands of atoms. In such contexts, favorable asymptotic scaling is essential, and warrants careful engineering of the model inductive bias.
>
> Various approaches exist that capture local chemical effects as highlighted by the reviewer, including MACE, Allegro (based off NequIP), PaiNN, etc. However, while the specifics of construction of local chemical embeddings vary between these approaches, they face a fundamental limitation in their inductive bias: **they do not incorporate long-range information**. For this reason, they are **inapplicable to our task**, where local material properties vary strongly due to confinement effects hundreds of atoms away.
>
> The novelty of our work lies in explicitly introducing boundaries into local atomic networks, and introducing a framework to communicate requisite long-range information to and from atomic nodes via equivariant attention mechanisms to capture geometric confinement. Crucially, we do this in a manner preserving O(N) asymptotic scaling, and demonstrate high accuracy and extensibility to planar nanosheets, including in OOD situations. We note that the choice of DOSTransformer to compare against was because this architecture also included linearly-scaling global graph interaction, albeit with a structure-agnostic cross-attention mechanism and inability to make local predictions.

---

> > ### Author Response · Authors · 2025-12-03
> >
> > **Question 2**
> > > "What are the atom counts in each nanoparticle?"
> >
> > Atom counts in the nanosheet very between 15 to 100 atoms, with the largest number of atoms in the <100> cleaving direction. Atoms in the nanowire dataset scale as the square of the critical dimension (as the cross section is a square), between 100-900 atoms.
> >
> > **Question 3/4/5**
> > > "How does EBFormer trained on TB perform on a DFT-labeled subset (no simulator leakage)? Results on III-V (GaAs, InP), wide-bandgap (GaN, SiC), oxides/heterostructures? Can you extend to band structures/effective masses, phonons/elastic moduli, dielectric spectra to stress different physical regimes? Any transfer learning experiments?"
> >
> > Due to the high computational cost of simulating our systems using DFT, we as of yet do not have a quantification of the performance of the network when using DFT compared to empirical TB. Similarly, data are not included in our custom dataset for various other material systems and properties. Generation of the dataset in its current form required ~4 months of computational wall time on three compute clusters with 112 processes hosted on Intel Xenon Gold 6330 processors.
> >
> > We have provided a transfer learning experiment in Appendix I "TRANSFER LEARNING" in our original submission, which quantified the improvement in OOD performance in Silicon nanosheets given large dimension Germanium nanosheet data, and notice marked improvement, indicating that EBFormer is able to leverage geometric information using spatial variation of atom-local quantities to improve global and local predictions.

---

### Meta-Review · Area_Chair_5ued · 2026-01-05

**Summary:**

The paper introduces a new method (EBFormer) for predicting properties of nanoscale devices using equivariant attention that includes global geometric features, such as the effect of boundaries in materials systems of nanoscale devices. The proposed architecture includes atomic level information as well information derived from boundaries that get processed into interaction blocks containing convolution and attention operations based on equivariant formulations. The paper applies its proposed model on a dataset of nanosheet and nanowire systems whose labels are generated by computational chemistry calculations by the authors themselves. The paper presents a set of experiments on the proposed systems including proposed in-distribution and out-of-distribution settings for the nanosheets and nanowires  comparing the proposed EBFormer, including some ablations of the method, across a set of baselines.

The reviewers generally praise the papers clarity of communicating the method and appreciate the proposed architecture that is motivated by the specific domain proposed in the paper. Some reviewers also highlight the efforts by the authors to create their own dataset.

The main concerns expressed by the reviewers relate to the applicability of the method beyond the specific cases described in the paper, prompting worries about the applicability of ICLR as a venue of publication. This is shown by concerns about benchmarking on common datasets as well as broader benchmarking and dataset construction to better motivate the applicability and limitations of EBFormer. Some reviewers also asked clarifying questions about the dataset generation, which is also tied to the applicability of the method. Overall, the paper would benefit from including the feedback of the reviewers to better motivate the design case of nanostructures from a machine learning modeling perspective, provide a deeper analysis of limitations of EBFormer that potentially includes benchmarking on other datasets or broadening the proposed dataset to show broader capabilities.

**Reviewer Concerns:**

Addressed Concerns:
* Reviewer GH4c's clarifications on the dataset construction, including number of atoms in different systems and some clarifications about dataset construction and task choice.
* Reviewer 8WJe's clarifications on experimental design and some scaling properties.
* Reviewer keTe's concerns related to novelty of the method and clarifications on how EBFormer handles certain cases.
* Reviewer fHnK's clarifications on some experimental details.

Outstanding Concerns:
* Reviewer GH4c's concerns on broader benchmarking. Some of the questions asked were not addressed during the author-reviewer discussion.
* Reviewer 8WJe's concerns on benchmarking and understanding capabilities and limitations of EBFormer. Some of the questions asked were not addressed during the author-reviewer discussion.
* Reviewer keTe's on understanding broader limitations of EBFormer.
* Reviewer fHnK's concerns on applicability of the method.

**Reviewer Scores:**

* Reviewer GH4c maintains score of 0, potentially raises to 2.
* Reviewer 8WJe maintains score of 6.
* Reviewer keTe maintains score of 2.
* Reviewer fHnK maintains score of 4.

---

### Decision · Program_Chairs · 2026-01-26

Reject